

# Conformal Carroll scalars with boosts

Stefano Baiguera[1], Gerben Oling[2], Watse Sybesma[3] and Benjamin T. Søgaard[4]

**1** Department of Physics, Ben-Gurion University of the Negev, Beer Sheva 84105, Israel
**2** Nordita, KTH Royal Institute of Technology and Stockholm University,
Hannes Alfvéns väg 12, SE-106 91 Stockholm, Sweden
**3** Science Institute, University of Iceland, Dunhaga 3, 107 Reykjavík, Iceland
**4** Department of Physics, Princeton University, Princeton, NJ 08544, USA

## Abstract

We construct two distinct actions for scalar fields that are invariant under local Carroll boosts and Weyl transformations. Conformal Carroll field theories were recently argued to be related to the celestial holography description of asymptotically flat spacetimes. However, only few explicit examples of such theories are known, and they lack local Carroll boost symmetry on a generic curved background. We derive two types of conformal Carroll scalar actions with boost symmetry on a curved background in any dimension and compute their energy-momentum tensors, which are traceless. In the first type of theories, time derivatives dominate and spatial derivatives are suppressed. In the second type, spatial derivatives dominate, and constraints are present to ensure local boost invariance. By integrating out these constraints, we show that the spatial conformal Carroll theories can be reduced to lower-dimensional Euclidean CFTs, which is reminiscent of the embedding space construction.



# 1 Introduction

Theories with Galilean symmetries can successfully describe many problems in astrophysics and classical mechanics, despite features such as absolute time and instantaneous interactions in for example Newton's laws that we ultimately believe to be unphysical. From an algebraic perspective, this is related to the observation that Galilean symmetries can be recovered from a non-relativistic $c \to \infty$ limit of the fundamental Lorentz symmetries. In this work, we are interested in the opposite $c \to 0$ limit, which leads to the so-called Carroll symmetries [1,2]. Despite some of their counter-intuitive features, such as ultra-local behavior, it turns out that theories with Carroll symmetries can nevertheless provide a worthwhile vantage point for describing physical theories.

One particular application of Carroll symmetries that we want to highlight arises in the context of flat space holography [3–7]. Here, the Bondi, van der Burg, Metzner, Sachs (BMS) algebra [8–13] captures the asymptotic symmetries that arise at null infinity. For this reason, the BMS$_d$ algebra associated to a $d$-dimensional asymptotically flat spacetime should be reflected in a putative $(d-1)$-dimensional holographic dual. It turns out that the BMS$_d$ algebra is isomorphic to the conformal extension of the Carroll group in $(d-1)$ dimensions [12],

$$\text{conformal Carroll}_{d-1} \simeq \text{BMS}_d \,. \tag{1.1}$$

Several works have explored the implication of this identification. A complementary approach to the holographic description of four-dimensional asymptotically flat spacetimes involves two-dimensional celestial conformal field theories (CCFT) on the celestial sphere [14–16]. In this approach, S-matrix elements, which are presumed to be the only gauge-invariant observables in quantum gravity and the only available observables at null infinity, are related to conformal correlation functions.

Several properties of such CCFTs are required to be significantly different than usual two-dimensional conformal field theory, and little is known about concrete theories implementing all these requirements. Recently, it has been argued that conformal Carroll theories can be related explicitly to CCFTs [17,18]. However, only few examples of theories realizing conformal Carroll symmetries exist, especially on the level of actions. As a result, a basic testing ground for more advanced concepts in conformal Carroll field theory and its connection to flat space holography is currently missing.

Even in the absence of conformal invariance, it has been a difficult task to determine Carroll field theories starting from a well-defined action principle. Several examples have been proposed, including scalars, fermions and gauge theories [19–22].

In the conformal case, two-dimensional scalar actions were obtained in [23]. In general dimensions, conformal scalar actions were derived using Carroll Weyl covariance in [24] and also recently in [25]. However, as we will explain in the main part of the paper, these theories are not required to be invariant under local Carroll boost symmetries, which we believe to be an essential part of the geometry. Without these local boost symmetries, the notion of

energy-momentum tensor appears to be problematic. For example, Killing vector fields do not lead to conserved currents without further conditions [26]. We will discuss this in more detail in Section 2.6 below. For this reason, we reconsider the conformal coupling of scalars to curved Carroll geometry. Using local Carroll boost symmetries, we can recover a regular energy-momentum tensor and also discover several new features of the resulting theories.

In flat space, Carroll boosts act on the time and spatial coordinates as

$$t \to t + \vec{v} \cdot \vec{x}, \qquad \vec{x} \to \vec{x}, \tag{1.2}$$

where $\vec{v}$ is a boost parameter. These boosts impose restrictions on the scalar two-point functions, which have two distinct classes of solutions [18, 22]

$$\langle \phi(t,x)\phi(0,0) \rangle = \begin{cases} f(t)\delta(x), \\ g(x). \end{cases} \tag{1.3}$$

These two distinct branches of solutions correspond to two different types of Carroll-invariant actions, which we will refer to as the *timelike* and *spacelike* theories [21, 22, 27, 28]. In the context of electromagnetism, they are usually known as *electric* and *magnetic* limit, depending on whether the gauge field is timelike or spacelike, respectively.[1]

To derive conformal Carroll scalar actions, we apply a covariant expansion in the speed of light around $c \to 0$ to a $d$-dimensional relativistic conformally coupled scalar action in curved spacetime. This was recently developed in the context of the ultra-local Carroll expansion of general relativity [28], building on related work for the non-relativistic expansion [29, 30]. For the spacelike action, we find that it is possible to dimensionally reduce to a relativistic Euclidean CFT in one lower dimension. Furthermore, we verify that their energy flux vanishes, as required by the local Carroll boost Ward identity [22, 31].

Conformal Carroll correlators are conjectured to be holographically related to observables in asymptotically flat spacetimes through a combination of Fourier and Mellin transformations [17] or through a modified Mellin transformation [18]. We will not pursue this line of thought in the current work, but the explicit theories we construct could for example be used as playgrounds to test general bootstrap techniques for BMS-invariant field theories following [32, 33] and the recent discussion of representations of the conformal Carroll algebra in [34]. Beyond the scope of flat space holography, Carroll symmetry can be viewed as a novel starting point for a perturbative study of interesting relativistic phenomena. In this sense, it is relevant for ultra-local limits of general relativity [28] and for the description of tensionless and null strings [35–40]. Furthermore, the Carrollian point of view provides new insights on fracton physics [41, 42], Love numbers [43], the membrane paradigm [44], the fluid/gravity correspondence [26, 45–50] and a possible road to understanding dark energy [22].

This paper is organized as follows. We review the essential features of Carroll geometry in Section 2, both from an intrinsic perspective and from a limit of Lorentzian geometry. Furthermore, we discuss the consequences on the energy-momentum tensor of Weyl and boost invariance and we compare this to other approaches to Carroll geometry and Carroll fluids (as summarized in [26]) in Section 2.6. In Sections 3 and 4 we develop two distinct Carroll limits of a relativistic theory coupled to Lorentzian geometry to find the timelike and spacelike Carroll-invariant actions, respectively. For each of them, we explicitly show their invariance under Carroll boosts and Weyl transformations and we derive the energy-momentum tensor using a variation of the background geometry. In the case of the spacelike action, we show in Section 4.3 that it is possible to dimensionally reduce to a relativistic Euclidean CFT in one lower dimension. Finally, we conclude in Section 5, where we also discuss possible future directions.

---

[1]Since the terminology *timelike* or *spacelike* is independent of the specialization to an electromagnetic theory, we will use this terminology throughout all the present work.

## 2 Carroll geometry background

In this section, we introduce the curved Carroll geometry that our scalar field theories will couple to. Several of its key ingredients have appeared previously in the literature, including its metric data, an appropriate connection and its curvature, and the notion of local Carroll boosts. We briefly review these ingredients in Sections 2.1 and 2.2, mainly following the discussion in [28]. There, they were also derived using an expansion of Lorentzian geometry in the speed of light, which we summarize in Section 2.3.

We then discuss Weyl transformations in Section 2.4. Next, in Section 2.5, we introduce a procedure for defining an energy-momentum tensor from a Carroll-invariant action using variations of the background Carroll geometry, and we discuss the corresponding Ward identities and conserved currents associated to isometries. Here, it is important to distinguish the notion of Carroll geometry considered here from other approaches in the literature, as recently summarized in [26]. We discuss their differences and a proposal for an identification of their geometric data in Section 2.6.

### 2.1 Metric data and boosts

The metric data of Carroll geometry consist of a vector field $v^\mu$ that designates the direction of time and a degenerate symmetric two-tensor $h_{\mu\nu}$ that, roughly speaking, acts as the metric for spatial directions. We can supplement these with a one-form $\tau_\mu$ and a two-tensor $h^{\mu\nu}$ so that

$$v^\mu h_{\mu\nu} = 0\,, \quad \tau_\mu h^{\mu\nu} = 0\,, \qquad v^\mu \tau_\mu = -1\,, \quad h^{\mu\rho}h_{\rho\nu} = \delta^\mu_\nu + v^\mu \tau_\nu\,. \tag{2.1}$$

We will conveniently denote the spatial projector in the last identity as $h^\mu_\nu \equiv h^{\mu\rho}h_{\rho\nu}$. The previous relations allow us to define a vielbein determinant as $e = \sqrt{\det(\tau_\mu \tau_\nu + h_{\mu\nu})}$. Its variation can be expressed as

$$\delta e = e\left(-v^\mu \delta\tau_\mu + \frac{1}{2}h^{\mu\nu}\delta h_{\mu\nu}\right) = e\left(\tau_\mu \delta v^\mu - \frac{1}{2}h_{\mu\nu}\delta h^{\mu\nu}\right). \tag{2.2}$$

Just as in Riemannian geometry, we can use $e$ to integrate scalar functions.

**Carroll boosts** In a general Carroll manifold, the flat space Carroll boosts (1.2) from the Introduction are generalized to the local Carroll boosts

$$\delta_\lambda v^\mu = 0\,, \quad \delta_\lambda h^{\mu\nu} = h^{\mu\rho}\lambda_\rho v^\nu + h^{\nu\rho}\lambda_\rho v^\mu\,, \qquad \delta_\lambda \tau_\mu = \lambda_\mu\,, \quad \delta_\lambda h_{\mu\nu} = 0\,. \tag{2.3}$$

Here, $\lambda_\mu(x)$ is a spacetime-dependent one-form which parametrizes the local boost velocity. It is 'spatial', in the sense that it satisfies $v^\mu \lambda_\mu(x) = 0$. As discussed in [28], Carroll boosts arise from an ultra-local $c \to 0$ limit of local Lorentz boosts. They are therefore an indispensable part of Carroll geometry and can be understood from an ultra-local expansion of Lorentzian geometry.

From (2.3) we see that not all of the metric variables of Carroll geometry are invariant under local boost transformations, in contrast to Lorentzian geometry. However, the physical quantities we consider will always be invariant under Carroll boosts, even though some of their terms may transform individually. For example, we can see that the vielbein determinant $e$ is invariant under local boosts, even though it involves the non-invariant one-form $\tau_\mu$, by plugging the boost transformations (2.3) into (2.2).

Since $v^\mu$ and $h_{\mu\nu}$ are boost-invariant, we can consider them as the fundamental metric quantities of a Carroll geometry. However, they do not uniquely determine the inverse quantities $\tau_\mu$ and $h^{\mu\nu}$. In fact, the ambiguity in solving (2.1) precisely corresponds to the boost

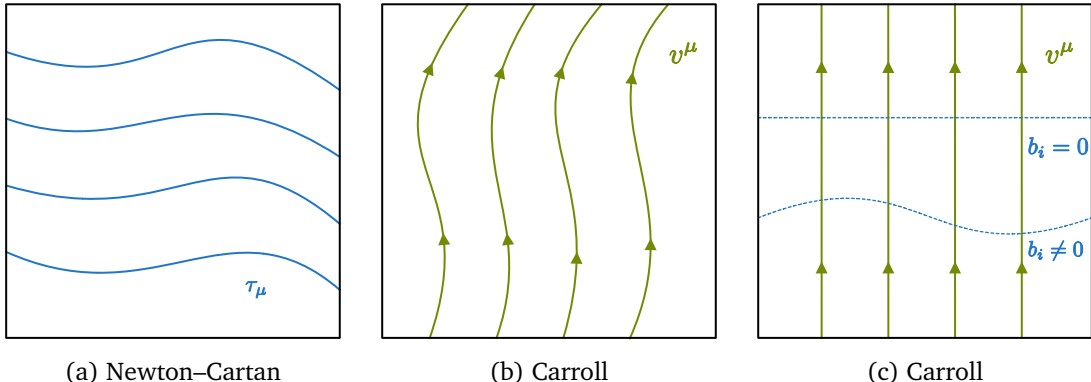

| (a) Newton–Cartan | (b) Carroll | (c) Carroll |

Figure 1: Foliation of spacetime in terms of a) spatial slices from $\tau_\mu$ in Newton–Cartan geometry, and b) timelike curves from $v^\mu$ in Carroll geometry. In the latter case, using adapted coordinates as in c), we can still use $\tau_\mu$ to define coordinates in a fixed boost frame.

freedom (2.3). On the other hand, we can uniquely solve the pair $\tau_\mu$ and $h_{\mu\nu}$ from the pair $v^\mu$ and $h^{\mu\nu}$, and vice versa. For this reason, we express metric variations in terms of these combinations, as in Equation (2.2).

**Spacetime foliations** In Newton–Cartan geometry, the analog of (2.3) are local Galilean boosts, which transform $v^\mu$ but leave $\tau_\mu$ invariant. As a result, it is natural in the context of local Galilean symmetry to define 'spacelike' vectors $X^\mu$ using the requirement $\tau_\mu X^\mu = 0$. If the one-form $\tau_\mu$ satisfies the Frobenius condition

$$\tau \wedge d\tau = 0 \,, \tag{2.4}$$

we can use it to define hypersurfaces whose tangent vectors are spacelike. A Galilean structure therefore naturally foliates spacetime into spatial slices, see Figure 1a.

In Carroll geometry, the nowhere-vanishing vector field $v^\mu$ is invariant under local boosts. Its integral curves then naturally lead to a foliation of a Carroll spacetime in one-dimensional 'timelike' curves, as depicted in Figure 1b, and $v^\mu$ can also be used to define 'spacelike' covectors $Y_\mu$ as those satisfying $v^\mu Y_\mu = 0$. We will occasionally raise the indices of spatial covectors using $Y^\mu = h^{\mu\nu} Y_\nu$, which can subsequently be lowered without loss of information due to (2.1).

On the other hand, since $\tau_\mu$ transforms under the local Carroll boosts (2.3), we do not have a natural notion of spacelike hypersurfaces as in Newton–Cartan geometry. Instead, *using* the local Carroll boost transformation of $\tau_\mu$, we can choose a boost frame such that (2.4) holds, as argued in [28]. In adapted coordinates $x^\mu = (t, x^i)$ with

$$v^\mu \partial_\mu = \partial_t \,, \qquad \tau_\mu dx^\mu = -dt + b_i dx^i \,, \tag{2.5}$$

we see that the boosts $\lambda_\mu dx^\mu = \lambda_i dx^i$ act on $\tau_\mu$ by shifting $b_i \to b_i + \lambda_i$. We can then use the boost symmetry to ensure (2.4) holds, for example by setting $b_i = 0$, so that the resulting $\tau_\mu$ can be used to define spatial slices, as in Figure 1c. It is important to realize that the choice of adapted coordinates (2.5) is non-unique and the above procedure of setting $b_i = 0$ does not lead to a unique $\tau_\mu$ given a set of Carroll data $(v^\mu, h_{\mu\nu})$. While the resulting spatial foliation depends on the boost and coordinate frame, it will still be useful in Section 4.3 below.

We will frequently do computations on the flat Carroll background, which is characterized by the following metric data:

$$v^\mu \partial_\mu = \partial_t \,, \quad \tau_\mu dx^\mu = -dt \,, \quad h_{\mu\nu} dx^\mu dx^\nu = \delta_{ij} dx^i dx^j \,, \quad h^{\mu\nu} \partial_\mu \partial_\nu = \delta^{ij} \partial_i \partial_j \,. \tag{2.6}$$

Note that $\tau_\mu$ and $h^{\mu\nu}$ are specified in a particular boost frame (corresponding to $b_i = 0$ above). They still transform under boosts, but this choice will be a useful way to define boost-invariant results.

**Extrinsic curvature and acceleration**  We will also frequently encounter

$$K_{\mu\nu} = -\frac{1}{2}\mathcal{L}_\nu h_{\mu\nu}, \tag{2.7}$$

which is the extrinsic curvature associated to $h_{\mu\nu}$. This symmetric two-tensor is spacelike, since $v^\mu K_{\mu\nu} = 0$. For this reason, we can define its trace as $K = h^{\mu\nu}K_{\mu\nu}$. Note that both $K_{\mu\nu}$ and $K$ are invariant under boosts. Finally, it is often useful to denote the exterior derivative of $\tau_\mu$ as $\tau_{\mu\nu} = 2\partial_{[\mu}\tau_{\nu]}$, and

$$a_\mu = v^\nu \tau_{\nu\mu} \tag{2.8}$$

is known as the acceleration. Since $v^\mu a_\mu = v^\mu v^\nu \tau_{\mu\nu} = 0$, this one-form is also spatial.

## 2.2  Connection and torsion

It is convenient to introduce a connection that is adapted to the Carroll metric quantities, and one possible choice is [28, 51, 52]

$$\tilde{\Gamma}^\rho_{\mu\nu} = -v^\rho \partial_{(\mu}\tau_{\nu)} - v^\rho \tau_{(\mu}\mathcal{L}_\nu\tau_{\nu)} + \frac{1}{2}h^{\rho\sigma}\left(\partial_\mu h_{\nu\sigma} + \partial_\nu h_{\sigma\mu} - \partial_\sigma h_{\mu\nu}\right) - h^{\rho\sigma}\tau_\nu K_{\mu\sigma}. \tag{2.9}$$

This connection is chosen such that the boost-invariant variables $v^\mu$ and $h_{\mu\nu}$ are covariantly constant with respect to the covariant derivative $\tilde{\nabla}$ associated to this connection. Conversely, $\tilde{\nabla}_\mu\tau_\nu$ and $\tilde{\nabla}_\rho h^{\mu\nu}$ are nonzero, but we will not need their explicit expressions here. This covariant derivative is not invariant under Carroll boosts, so in physical quantities it will only appear through boost-invariant combinations. We will explicitly check the boost invariance of the actions involving curvature that we construct below.

The connection (2.9) has nonzero torsion,

$$\tilde{T}^\rho{}_{\mu\nu} = 2\tilde{\Gamma}^\rho_{[\mu\nu]} = 2h^{\rho\lambda}\tau_{[\mu}K_{\nu]\lambda}. \tag{2.10}$$

We can write a total covariant derivate as

$$\tilde{\nabla}_\mu X^\mu = \partial_\mu X^\mu + \tilde{\Gamma}^\rho_{\rho\nu}X^\nu = \frac{1}{e}\partial_\mu(eX^\mu) - K\tau_\mu X^\mu, \tag{2.11}$$

which allows us to do integration by parts. We will frequently drop boundary terms, which we denote by a "$\approx$" symbol. In the following, we will also want to do integration by parts for Lie derivatives $\mathcal{L}_\nu$ with respect to $v^\mu$. Using $\mathcal{L}_\nu e = -eK$, which follows from the general variation (2.2), we see that

$$\int\left(v^\mu\partial_\mu f\right)g\,e\,d^dx = \int(\mathcal{L}_\nu f)\,g\,e\,d^dx \approx \int\left[-f\left(v^\mu\partial_\mu g\right) + fgK\right]e\,d^dx. \tag{2.12}$$

Note that this identity does not depend on the connection.

## 2.3  From an expansion of Lorentzian geometry

Starting from a Lorentzian geometry, we can obtain a Carroll geometry using the following procedure [28], which is similar to the procedure that was developed for the non-relativistic Galilean or Newton–Cartan expansion in [30].

First, we reparametrize the metric and its inverse using variables that designate a particular 'time' direction,

$$g_{\mu\nu} = -c^2 T_\mu T_\nu + \Pi_{\mu\nu}, \qquad g^{\mu\nu} = -\frac{1}{c^2} V^\mu V^\nu + \Pi^{\mu\nu}, \qquad (2.13)$$

$$T_\mu V^\mu = -1, \quad T_\mu \Pi^{\mu\nu} = 0, \qquad V^\mu \Pi_{\mu\nu} = 0, \quad \Pi^{\mu\rho} \Pi_{\rho\nu} = \delta^\mu_\nu + V^\mu T_\nu. \qquad (2.14)$$

This parametrization, which is similar to the $3 + 1$ decomposition that is used in the ADM formalism, allows us to factor out overall powers of the speed of light $c$ in the geometry. As such, it prepares the geometry for a uniform expansion around the ultra-local $c \to 0$ Carroll limit, and so it is also known as the 'pre-ultra-local' (PUL) parametrization.

The parametrization (2.13) singles out a time direction, since it is not invariant under local Lorentz transformations. Given such a choice, we can obtain Carroll metric variables from the leading-order terms in a $c \to 0$ expansion,

$$V^\mu = v^\mu + \mathcal{O}(c^2), \quad \Pi_{\mu\nu} = h_{\mu\nu} + \mathcal{O}(c^2), \quad T_\mu = \tau_\mu + \mathcal{O}(c^2), \quad \Pi^{\mu\nu} = h^{\mu\nu} + \mathcal{O}(c^2). \quad (2.15)$$

Higher-order terms will give corrections to the leading-order Carroll geometry, but we will not make use of them here. The expansion of the local Lorentz transformations $\Lambda^a{}_b(x)$ acting on local frames $E^a{}_\mu(x)$ reproduces the local Carroll boosts (2.3) at leading order. Algebraically, this corresponds to the İnönü–Wigner contraction of the Poincaré algebra to the Carroll algebra that was originally introduced by Lévy-Leblond [1].

Note that we have assumed the variables to be analytic in $c^2$. Performing a coordinate transformation in the Lorentzian geometry that breaks this assumption prior to the expansion will generically lead to an inequivalent Carroll geometry. Likewise, performing local Lorentz boosts prior to the expansion leads to a different Carroll geometry, since they change the designated time direction.

The torsionful Carroll connection (2.9) that we introduced previously can be obtained from the Levi-Civita connection as follows. First, using the parametrization (2.13) we can rewrite the latter as

$$\Gamma^\rho_{\mu\nu} = \frac{1}{c^2} \overset{(-2)}{S}{}^\rho{}_{\mu\nu} + \overset{(0)}{S}{}^\rho{}_{\mu\nu} + \tilde{C}^\rho_{\mu\nu} + c^2 \overset{(2)}{S}{}^\rho{}_{\mu\nu}, \qquad (2.16)$$

where $\tilde{C}^\rho_{\mu\nu}$ is the PUL analogue of the Carroll connection $\tilde{\Gamma}^\rho_{\mu\nu}$ from (2.9),

$$\tilde{C}^\rho_{\mu\nu} = -V^\rho \partial_{(\mu} T_{\nu)} - V^\rho T_{(\mu} \mathcal{L}_V T_{\nu)} + \frac{1}{2} \Pi^{\rho\sigma} \left( \partial_\mu \Pi_{\nu\sigma} + \partial_\nu \Pi_{\sigma\mu} - \partial_\sigma \Pi_{\mu\nu} \right) - \Pi^{\rho\sigma} T_\nu \mathcal{K}_{\mu\sigma}. \quad (2.17)$$

Using the expansion (2.15), this reproduces the Carroll connection (2.9) at leading order. Here, we have defined

$$T_{\mu\nu} = 2\partial_{[\mu} T_{\nu]}, \qquad \mathcal{K}_{\mu\nu} = -\frac{1}{2} \mathcal{L}_V \Pi_{\mu\nu}, \qquad (2.18)$$

in analogy with the previous subsection. The remaining tensors in (2.16) are

$$\overset{(-2)}{S}{}^\rho{}_{\mu\nu} = -V^\rho \mathcal{K}_{\mu\nu}, \qquad \overset{(0)}{S}{}^\rho{}_{\mu\nu} = \Pi^{\rho\lambda} T_\nu \mathcal{K}_{\mu\lambda}, \qquad \overset{(2)}{S}{}^\rho{}_{\mu\nu} = -T_{(\mu} \Pi^{\rho\lambda} T_{\nu)\lambda}. \quad (2.19)$$

Following the decomposition (2.16) of the Levi-Civita connection, we can rewrite the Lorentzian Levi-Civita Ricci scalar as follows,

$$R = \frac{1}{c^2} \left[ \mathcal{K}^{\mu\nu} \mathcal{K}_{\mu\nu} + \mathcal{K}^2 - 2V^\mu \partial_\mu \mathcal{K} \right] + \Pi^{\mu\nu} \overset{(C)}{R}{}_{\mu\nu} + \frac{c^2}{4} \Pi^{\mu\nu} \Pi^{\rho\sigma} T_{\mu\rho} T_{\nu\sigma} - \overset{(C)}{\nabla}_\mu (V^\nu \Pi^{\mu\rho} T_{\nu\rho}), \quad (2.20)$$

where $\overset{(C)}{R}{}_{\mu\nu}$ is the Ricci tensor of the connection (2.17). Here, $\mathcal{K} = \Pi^{\mu\nu} \mathcal{K}_{\mu\nu}$ denotes the trace of the extrinsic curvature. Finally, we can integrate by parts using the identity

$$\overset{(C)}{\nabla}_\mu X^\mu = \partial_\mu X^\mu + \overset{(C)}{\Gamma}{}^\rho_{\rho\nu} X^\nu = \frac{1}{E} \partial_\mu (E X^\mu) - \mathcal{K} T_\mu X^\mu. \qquad (2.21)$$

Here, $E = \sqrt{\det(T_\mu T_\nu + \Pi_{\mu\nu})} = \sqrt{-g}/c$ is the Lorentzian vielbein determinant, which reduces to the Carroll vielbein determinant $e$ at leading order in the ultra-local expansion.

## 2.4 Weyl transformations

In Lorentzian geometry, we can parametrize infinitesimal Weyl transformations of the metric as follows,

$$\delta_\omega g_{\mu\nu} = 2\omega g_{\mu\nu}, \qquad \delta_\omega g^{\mu\nu} = -2\omega g^{\mu\nu}. \tag{2.22}$$

In the context of Carroll geometry, we consider the Weyl transformations[2]

$$\delta_\omega v^\mu = -\omega v^\mu, \qquad \delta_\omega \tau_\mu = \omega \tau_\mu, \qquad \delta_\omega h^{\mu\nu} = -2\omega h^{\mu\nu}, \qquad \delta_\omega h_{\mu\nu} = 2\omega h_{\mu\nu}. \tag{2.23}$$

These transformations are compatible with the Lorentzian transformations (2.22) after the pre-ultra-local parametrization (2.13) and the expansion (2.15).

Under the Weyl transformations (2.23), the extrinsic curvature (2.7), the acceleration one-form (2.8) and the Carroll-compatible connection (2.9) transform as

$$\delta_\omega K_{\mu\nu} = \omega K_{\mu\nu} - h_{\mu\nu} v^\rho \partial_\rho \omega, \qquad \delta_\omega K = -\omega K - (d-1) v^\rho \partial_\rho \omega, \tag{2.24}$$

$$\delta_\omega a_\mu = -2 h_{\mu\nu} h^{\nu\rho} \partial_\rho \omega, \tag{2.25}$$

$$\delta \tilde{\Gamma}^\rho_{\mu\nu} = \delta^\rho_\mu \partial_\nu \omega + \delta^\rho_\nu \partial_\mu \omega - h_{\mu\nu} h^{\rho\lambda}(\partial_\lambda \omega) + \delta^\rho_\mu \tau_\nu v^\sigma (\partial_\sigma \omega). \tag{2.26}$$

Finally, following (2.2), the Carroll vielbein determinant transforms as

$$\delta e = d\omega\, e. \tag{2.27}$$

The actions we construct will be invariant under these Carroll Weyl transformations.

## 2.5 Energy-momentum tensor and Ward identities

Similar to the relativistic case, we can define an energy-momentum tensor for a Carroll-invariant action by varying with respect to the background Carroll geometry. First, varying the action gives rise to two (sets of) currents,

$$\delta S = -\int d^d x\, e \left[ T^\nu_\mu \delta v^\mu + \frac{1}{2} T^h_{\mu\nu} \delta h^{\mu\nu} \right]. \tag{2.28}$$

Note that we vary the action with respect to $v^\mu$ and $h^{\mu\nu}$ following the discussion below the boost transformations (2.3). If the action $S$ is invariant under local boosts, it follows from the transformations of the metric variables that $T^\nu_\mu$ transforms under boosts,

$$\delta_\lambda T^\nu_\mu = -T^h_{\mu\nu} \lambda^\nu, \quad \delta_\lambda T^h_{\mu\nu} = 0, \tag{2.29}$$

which cancels the transformation of the metric variation $\delta h^{\mu\nu}$ under boosts. We can then combine these two currents into a single energy-momentum tensor [56,57]

$$T^\mu{}_\nu = -v^\mu T^\nu_\nu - h^{\mu\rho} T^h_{\rho\nu}, \tag{2.30}$$

which is invariant under local Carroll boost. We will discuss in Section 2.6 below how this notion of Carroll energy-momentum tensor and the following properties are related to other constructions in the literature.

---

[2]From an intrinsic Carroll perspective, it is possible to introduce an inhomogeneous Lifshitz scaling parameter $z$ between the 'time' and 'space' directions. This leads to a generalization of Equation (2.23), which was considered in, for example, [53–55]. Since we are mainly interested in Carroll theories that descend from limits of Lorentzian theories, we restrict ourselves to the uniform $z = 1$ scaling. It is also only for this value of $z$ that the isomorphism (1.1) between the conformal Carroll algebra and the BMS algebra holds [12, 13].

**Boost Ward identity**    If we specialize the metric variations in (2.28) to boost transformations, invariance of the action under arbitrary local boosts implies

$$T^h_{\mu\nu}\lambda^\mu v^\nu = 0 \quad \implies \quad T^h_{\rho\nu}v^\nu = 0 \quad \implies \quad h_{\mu\rho}T^\rho{}_\nu v^\nu = 0\,, \tag{2.31}$$

which holds without using the equations of motion. It is important to stress that this Ward identity follows from the invariance under local Carroll boosts. In particular, it is unrelated to diffeomorphism symmetry.

**Diffeomorphism Ward identity**    If we act on an action $S$ with an infinitesimal coordinate transformation $x^\mu \to x^\mu + \xi^\mu(x)$, we get the on-shell variation

$$\begin{aligned}
\delta_\xi S &= \int d^d x\; e\left[-T^\nu_\mu \mathcal{L}_\xi v^\mu - \frac{1}{2}T^h_{\mu\nu}\mathcal{L}_\xi h^{\mu\nu}\right] \\
&= \int d^d x\; e\xi^\rho\left[-T^\nu_\mu \partial_\rho v^\mu - \frac{1}{2}T^h_{\mu\nu}\partial_\rho h^{\mu\nu} + e^{-1}\partial_\nu\left[e\left(-T^\nu_\rho v^\nu - T^h_{\rho\alpha}h^{\alpha\nu}\right)\right]\right].
\end{aligned} \tag{2.32}$$

As a result, diffeomorphism invariance of $S$ then implies

$$e^{-1}\partial_\nu\left[eT^\nu{}_\rho\right] = T^\nu_\mu \partial_\rho v^\mu + \frac{1}{2}T^h_{\mu\nu}\partial_\rho h^{\mu\nu}\,. \tag{2.33}$$

On the flat background (2.6), this reduces to the conservation of the energy-momentum tensor. This identity is not covariant, however we will use it as an intermediate result to establish a covariant derivation for the conservation of the current $T^\mu{}_\nu\xi^\nu$ below.

**Weyl Ward identity**    Additionally, as in a Lorentzian theory, Weyl invariance of the action implies that the energy-momentum tensor (2.30) is traceless. Specializing the metric variations of the action in (2.28) to the Weyl transformations (2.22), we get

$$\delta_\omega S = \int d^d x\; e\omega(x)T^\mu{}_\mu \quad \implies \quad T^\mu{}_\mu = -T^\nu_\mu v^\mu - T^h_{\mu\nu}h^{\mu\nu} = 0\,. \tag{2.34}$$

The tracelessness of the energy-momentum tensor holds upon using the equations of motion. In a quantum theory, this Ward identity may become anomalous.

**Conserved currents for isometries**    Finally, the diffeomorphism Ward identity (2.33) allows one to construct conserved currents associated to Carrollian isometries. A Carrollian Killing vector field $\xi^\mu$ is defined by preserving the Carrollian metric data,

$$\mathcal{L}_\xi v^\mu = 0\,, \qquad \mathcal{L}_\xi h_{\mu\nu} = 0\,. \tag{2.35}$$

For such a vector field, we can show that $T^\mu{}_\nu\xi^\nu$ is conserved, independent of a choice of connection. To do this, it is useful to dualize to the $(d-1)$-form

$$J_{\mu_1\dots\mu_{d-1}} = \xi^\rho T^\nu{}_\rho\, e\,\epsilon_{\nu\mu_1\dots\mu_{d-1}}\,. \tag{2.36}$$

Here, $\epsilon_{\mu_1\dots\mu_d}$ is the totally antisymmetric Levi-Civita symbol, which becomes a boost-invariant tensor when multiplied by the vielbein determinant. To demonstrate that the current $T^\mu{}_\nu\xi^\nu$ is conserved, we now show $J$ is closed. First, we can rewrite

$$dJ_{\mu_1\dots\mu_{d-1}} = \partial_\mu(e\xi^\nu T^\mu{}_\nu)\epsilon_{\mu_1\dots\mu_d}\,. \tag{2.37}$$

Then, using the diffeomorphism Ward identity (2.33), we see that

$$\partial_\mu(e\xi^\nu T^\mu{}_\nu) = e\left[T^\nu_\mu \mathcal{L}_\xi v^\mu + \frac{1}{2}T^h_{\mu\nu}\mathcal{L}_\xi h^{\mu\nu}\right] = eT^h_{\mu\nu}v^\mu h^{\nu\rho}\mathcal{L}_\xi \tau_\rho = 0. \tag{2.38}$$

In the second equality, we used the Killing equation (2.35) together with the identity $\delta h^{\mu\nu} = -h^{\mu\rho}h^{\nu\sigma}\delta h_{\rho\sigma} + 2v^{(\mu}h^{\nu)\rho}\delta\tau_\rho$, which follows from the orthonormality relations (2.1). The resulting expression vanishes since $T^h_{\mu\nu}v^\nu = 0$ due to the boost Ward identity (2.31). Hence, we conclude that $J$ is closed and that we have a conserved charge associated to each isometry. This construction generalizes to conformal isometries,

$$\mathcal{L}_\xi v^\mu = -\omega v^\mu, \qquad \mathcal{L}_\xi h_{\mu\nu} = 2\omega h_{\mu\nu}, \tag{2.39}$$

provided that the energy-momentum tensor is traceless, corresponding to the Ward identity (2.34) for Weyl transformations.

## 2.6 Comparison to other work

In this subsection we will comment on how the notion of Carroll geometry and the energy-momentum tensor discussed above relate to other constructions in the literature. This discussion is mostly independent from the rest of the paper, and a general reader that is not interested in this comparison can skip it without loss of continuity.

**Geometric structure**   The modern notion of a Carroll manifold was first introduced in [12, 13, 27] as a so-called 'weak' Carroll structure, consisting of a manifold $M$ endowed with a degenerate metric $h_{\mu\nu}$ whose kernel is spanned by a vector field $v^\mu$. This perspective was further developed in [51, 52], textitasizing the role of local Carroll boost as a gauge transformation arising from the non-canonical choice of $\tau_\mu$, and it was subsequently applied to gravity and field theory in for example [22, 28, 58, 59]. Any such Carroll theory is invariant under local boost transformations (2.3), which implement the Carrollian equivalence principle and removes the unphysical degrees of freedom that would otherwise be contained in the inverse tensors $h^{\mu\nu}$ and $\tau_\mu$.

In the literature there also exists another approach to curved Carroll spacetimes [24, 26, 46, 47, 60, 61], which is distinct at first sight. In this approach, an important role is played by the notion of *Carroll diffeomorphisms*. From our perspective, these arise as the diffeomorphisms that preserve the metric data $v^\mu$ and $h_{\mu\nu}$ in a specific class of adapted coordinate systems. Using the notation of [26], these correspond to $x^\mu = (t, x^i)$ such that[3]

$$v^\mu\partial_\mu = \Omega^{-1}\partial_t, \qquad h_{\mu\nu}dx^\mu dx^\nu = a_{ij}dx^i dx^j, \tag{2.40a}$$

$$\tau_\mu dx^\mu = -\Omega dt - b_i dx^i, \qquad h^{\mu\nu}\partial_\mu\partial_\nu = \Omega^{-2}b_i b^i \partial_t^2 + 2\Omega^{-1}b^i\partial_t\partial_i + a^{ij}\partial_i\partial_j, \tag{2.40b}$$

where the indices $i, j, \dots$ are raised using the matrix inverse $a^{ij}$ of the spatial metric $a_{ij}$. The Carroll diffeomorphisms are then the subset of coordinate transformations that preserve the form of $v^\mu$ and $h_{\mu\nu}$ in (2.40a) above.

**Local Carroll boosts**   In this alternative approach to Carroll geometry, the geometric data is taken to be $(\Omega, a_{ij}, b_i)$. Using the parametrization (2.40), this corresponds to the spatial metric $h_{\mu\nu}$ and vector field $v^\mu$ above. However, in contrast to our previous discussion, this choice of data means that the one-form $\tau_\mu$ and inverse spatial metric $h^{\mu\nu}$ are also fundamental Carroll objects.

---

[3]Note that we use an opposite sign in the contraction $v^\mu\tau_\mu = -1$ in comparison to [26] to be consistent with our conventions in (2.1) above.

This parametrization (2.40) of the geometry is covariant under the restricted set of coordinate transformations known as *Carroll diffeomorphisms* [46, 47]. On a flat background, these include global Carroll boosts, but this type of Carroll geometry does not require the existence of local boosts (2.3). If it would, $\tau_\mu$ would transform under them, which would be in conflict with considering $b_i$ as a fundamental object.

Many of our results crucially rely on local Carrollian boosts and the associated Ward identity (2.31). Therefore, several of our results are incompatible with for example the recent work [26], which provides a useful overview of this alternative notion of Carroll geometry and its consequences. Again, in our view, the main reason for these discrepancies comes down to the inclusion of local boost invariance. Additionally, we work with generally covariant quantities that are invariant under arbitrary diffeomorphisms instead of the subset of Carroll that preserves the parametrization (2.40), but we believe that this distinction is of secondary importance.

A simple example of this difference in the notion of Carroll-invariant theories is in coupling a massless (non-conformal) scalar $\phi$ to an arbitrary Carrollian background. Two candidate scalar actions are

$$S_t = \frac{1}{2} \int d^d x \, e v^\mu v^\nu \partial_\mu \phi \partial_\nu \phi = \frac{1}{2} \int d^d x \, \Omega \sqrt{a} \, (\Omega^{-1} \partial_t \phi)^2 \,, \tag{2.41a}$$

$$S_s = \frac{1}{2} \int d^d x \, e h^{\mu\nu} \partial_\mu \phi \partial_\nu \phi = \frac{1}{2} \int d^d x \, \Omega \sqrt{a} \, a^{ij} \hat{\partial}_i \phi \hat{\partial}_j \phi \,, \tag{2.41b}$$

where we have written the action both in covariant form as well as using the parameterization (2.40) with the definition $\hat{\partial}_i = \partial_i + \frac{b_i}{\Omega} \partial_t$ of [26]. The action $S_t$ is a Carrollian action in the sense used in this paper as the scalar couples only to the boost-invariant objects $e$ and $v^\mu$. On the other hand, $S_s$ does not enjoy local boost invariance and consequently its energy-momentum tensor does not automatically satisfy the boost Ward identity. The form of $S_s$ is invariant under Carrollian diffeomorphisms and thus it would be admissible as a Carrollian theory in the sense of [26]. Indeed, the derivative coupling in $S_s$ has appeared in papers using this alternative notion of Carrollian geometry such as [24, 25, 60].

**Energy-momentum tensor**   One place where the coordinate choice (2.40) initially has consequences is in computing the responses of the theory to variations of the metric data. In particular, in this parametrization the vector field $v^\mu$ is constrained to be proportional to $\partial_t$ and cannot probe the $\partial_i$ directions.

As a result, the time-space component $\tau_\mu T^\mu{}_\nu h^\nu{}_\rho$ of our energy-momentum tensor (2.30) cannot be computed by varying with respect to $(\Omega, a_{ij}, b_i)$. The authors of [26] do remark that such a momentum $P_i$ should exist after carefully analyzing the diffeomorphism Ward identity in the frame (2.40), but conclude that it is not conjugate to a geometric object. One can obtain the momentum $P_i$ by extending the parameterization (2.40) to using an additional field $f^i$ as follows,

$$v^\mu \partial_\mu = \Omega^{-1}(\partial_t + f^i \partial_i) \,, \qquad h_{\mu\nu} dx^\mu dx^\nu = f^i f_i dt^2 - 2 f_i dt dx^i + a_{ij} dx^i dx^j \,. \tag{2.42}$$

The extended parametrization of $h_{\mu\nu}$ is chosen such that $v^\mu h_{\mu\nu} = 0$ still holds. The restricted parametrization (2.40) corresponds to $f^i = 0$ above. The expressions for $\tau_\mu$ and $h^{\mu\nu}$ are modified in a similar way. With this, the missing momentum $P_i$ is now related to the response of varying with respect to $f^i$. As a result, we see that with this addition (2.42) now parametrizes a vector field $v^\mu$ and a spatial two-tensor $h_{\mu\nu}$ that are fully general. In Section 2.5, we have defined the energy-momentum tensor using arbitrary variations $\delta S/\delta v^\mu$ and $\delta S/\delta h^{\mu\nu}$, and our answers thus agree with the above once $f^i$ is included as in (2.42).

A separate question is the consequence of local boost invariance. As we already briefly mentioned below Equation (2.5), these boost transformations map to the transformations $b_i \to b_i + \lambda_i$ in such coordinates. For this reason, $b_i$ should have a vanishing conjugate energy or momentum in our formalism, and indeed one can show that the vanishing of $\delta S/\delta b_i$ corresponds precisely to the boost Ward identity (2.31). Conversely, since local Carroll boosts are not included in the formalism of [26], their energy-momentum tensor does not necessarily satisfy our Ward identity, see also [60]. However, we will now give two arguments for why we think it is natural to include local Carroll boosts.

**Killing vectors and conserved currents**  First, in the recent paper [26] the authors show that Carrollian Killing vectors (2.35) generically do not lead to an on-shell conservation law unless further constraints are imposed. This result appears to be in tension with our previous Equation (2.38), where we showed that we can define a conserved current $T^\mu{}_\nu \xi^\nu$ for any Killing vector field.

However, from our perspective, this discrepancy can be traced back to the absence of local boost invariance in [26], as the conservation of (2.38) depends crucially on our boost Ward identity (2.31). Indeed, without boost invariance, the aforementioned authors find that the non-conservation of their current is proportional to $\Pi^i \propto \delta S/\delta b_i$. This metric response vanishes precisely if local boost invariance is imposed. The apparent disagreement between our Equation (2.38) and [26] is therefore purely a question of the assumption of local Carroll boost invariance.

**Local boosts from a limit**  Second, we want to argue that local Carroll boosts arise naturally in the ultra-local $c \to 0$ limit of Lorentzian theories. For this, consider a Lorentzian action $S[g^{\mu\nu}, \phi, \dots]$ with a suitable overall factor[4] of $c^N$ such that the action is finite in the limit $c \to 0$, when written out in terms of the PUL parameterization (2.13),

$$g^{\mu\nu} = -\frac{1}{c^2} V^\mu V^\nu + \Pi^{\mu\nu}. \tag{2.43}$$

In fact, this over-parametrizes a generic metric $g^{\mu\nu}$ since $V^\mu$ designates a particular time direction, and such a choice is not contained in the Lorentzian metric data. This over-parameterization implies a relation between the functional derivatives of the action with respect to the PUL variables,

$$V^\mu \frac{\delta S}{\delta \Pi^{\mu\nu}} = -\frac{c^2}{2} \Pi^{\mu\rho} \Pi_{\rho\nu} \frac{\delta S}{\delta V^\mu}. \tag{2.44}$$

Then consider the scaling of these quantities in the $c \to 0$ limit. By assumption, both $\delta S/\delta V^\nu$ and $\delta S/\delta \Pi^{\mu\nu}$ are at most $\mathcal{O}(c^0)$. Hence, the relation above can only hold if the contraction $V^\nu(\delta S/\delta \Pi^{\mu\nu})$ actually is $\mathcal{O}(c^2)$. In the limit $c \to 0$, the latter is equivalent to the Carroll boost Ward identity (2.31). This argument shows that invariance under local Carroll boosts is automatically satisfied by Carroll theories that are obtained as the small $c$ limit of the PUL parametrization of a Lorentzian theory.

Note that the above does not depend on any frame bundle structure. Of course, one can also understand the appearance of local Carroll boosts more directly by considering the limit of local Lorentz boosts in the frame bundle, as described in [28].

---

[4]We can always include such an overall factor by performing an appropriate field redefinition.

**Summary** Before concluding this section, we summarize a number of properties of $T^\mu{}_\nu$ as defined in (2.30) that make it a natural candidate for a spacetime energy-momentum tensor of a Carroll-invariant field theory:

1. At least for scalar theories, $T^\mu{}_\nu$ coincides with the canonical energy-momentum tensor obtained by the Noether procedure (up to Belinfante improvement terms).

2. It allows for the construction of conserved currents associated to (conformal) Killing vector fields without any additional constraints on the geometry.

3. It encodes the full content of the response to varying the metric data $(v^\mu, h^{\mu\nu})$ in a boost-invariant manner.

In conclusion, some care is required in comparing results coming from the notions of Carroll geometry used in this paper and the one reviewed recently in [26]. Apart from some notational differences, the main distinction between these two notions is the existence of local Carroll boost invariance. As we argued above, this is a natural property of Carroll theories that are derived from a limit of Lorentzian theories, and it furthermore allows us to construct a spacetime energy-momentum tensor with very similar properties to its Lorentzian counterpart.

# 3 Timelike conformal Carroll scalar action

Having discussed the relevant aspects of Carroll geometry, we now turn to the main goal of this paper. Starting from the conformally coupled relativistic scalar, we will derive two distinct classes of scalar actions that are conformally coupled to Carroll backgrounds. We will refer to them as the 'timelike' and 'spacelike' actions due to the type of derivatives in their respective kinetic operators. We first focus on the timelike case.

Our starting point is the usual Lorentzian conformally coupled scalar action

$$S = -\frac{1}{2} \int d^d x \, \sqrt{-g} \left[ g^{\mu\nu} \partial_\mu \phi \, \partial_\nu \phi + \frac{(d-2)}{4(d-1)} R \phi^2 \right]. \tag{3.1}$$

This action contains the Lorentzian background metric $g_{\mu\nu}$ as well as the Ricci scalar $R$ of the corresponding Levi-Civita connection. It is invariant under the Lorentzian Weyl transformations (2.22).

Using the pre-ultra-local parametrization of the Lorentzian geometry that was introduced in Section 2.3, we can rewrite this action in a form that is suitable for a covariant Carroll expansion. In particular, rewriting the metric using (2.13) and the Levi-Civita Ricci scalar using (2.20), the action (3.1) becomes

$$S = -\frac{c^2}{2} \int d^d x \, E \left[ \left( -\frac{1}{c^2} V^\mu V^\nu + \Pi^{\mu\nu} \right) \partial_\mu \phi \, \partial_\nu \phi \right.$$
$$+ \frac{(d-2)}{4(d-1)} \left( \frac{1}{c^2} \left[ \mathcal{K}^{\mu\nu} \mathcal{K}_{\mu\nu} + \mathcal{K}^2 - 2 V^\mu \partial_\mu \mathcal{K} \right] + \Pi^{\mu\nu} \tilde{R}_{\mu\nu} \right. \tag{3.2}$$
$$\left. \left. - \tilde{\nabla}_\mu (V^\nu \Pi^{\mu\rho} T_{\nu\rho}) + \frac{c^2}{4} \Pi^{\mu\nu} \Pi^{\rho\sigma} T_{\mu\rho} T_{\nu\sigma} \right) \phi^2 \right].$$

This action still describes precisely the same dynamics as the usual action (3.1) above. In this expression, we have rescaled the fields using $\phi \to c^{1/2} \phi$ so that the leading-order term is finite in the ultra-local $c \to 0$ limit. Taking this limit, we obtain

$$S = -\frac{1}{2} \int d^d x \, e \left( -(v^\mu \partial_\mu \phi)^2 + \frac{(d-2)}{4(d-1)} \left[ K^{\mu\nu} K_{\mu\nu} + K^2 - 2 v^\rho \partial_\rho K \right] \phi^2 \right). \tag{3.3}$$

This action contains the Carroll geometric variables defined in Section 2.1. In this limit, time derivatives dominate and spatial derivatives are suppressed. For this reason, we refer to (3.3) as the timelike action.

The flat space version of (3.3) has been considered in several places, for example recently in [18,22]. In two dimensions, the conformal coupling vanishes, and the corresponding action in curved backgrounds has recently been discussed in [23]. In addition, [24,25] considered a similar action coupling to a curved background geometry using the alternative approach Carroll geometry that we discussed in Section 2.6. However, following our earlier discussion, the invariance under local Carroll boosts of the action (3.3) was not exploited in that works.

Since the action (3.3) is derived from the Lorentzian conformal scalar action (3.1), it should be invariant under local Carroll boosts and Weyl transformations, and we will show this explicitly in Section 3.1. Additionally, in Section 3.2 we derive the energy-momentum tensor of the action (3.3) by varying with respect to the Carroll background metric data. In the classical theory, this energy-momentum tensor is traceless and satisfies a Ward identity associated to Carroll boosts.

## 3.1 Carroll boost and Weyl symmetries

The boost invariance of the action (3.3) follows from the fact that all tensors entering in it are invariant under local Carroll boosts (2.3). First, recall that the timelike vector field $v^\mu$ and the spatial metric $h_{\mu\nu}$ are boost-invariant. The extrinsic curvature $K_{\mu\nu}$ corresponds to the Lie derivative of $h_{\mu\nu}$ with respect to $v^\mu$, so it is also boost-invariant.

To see the Weyl invariance of the action, we use the Carroll Weyl transformations discussed in Section 2.4. Together with the standard Weyl transformation of the scalar,

$$\delta_\omega \phi = -\frac{(d-2)}{2}\omega\phi\,, \tag{3.4}$$

we see that the conformal coupling term in the action (3.3) transforms as

$$\delta_\omega\left(-e\frac{(d-2)}{4(d-1)}\left[K^{\mu\nu}K_{\mu\nu}+K^2-2v^\rho\partial_\rho K\right]\phi^2\right) = e\frac{(d-2)}{2}K\left(v^\mu\partial_\mu\omega\right)\phi^2 - e\frac{(d-2)}{2}\left(\mathcal{L}_v\mathcal{L}_v\omega\right)\phi^2 \tag{3.5}$$

$$\approx e(d-2)\phi\left(v^\mu\partial_\mu\phi\right)\left(v^\nu\partial_\nu\omega\right)\,. \tag{3.6}$$

In the last line, we have used the Lie derivative identity (2.12) to integrate by parts, dropping total derivatives. Next, the Weyl transformation of the kinetic term gives

$$\delta_\omega\left[e\left(v^\mu\partial_\mu\phi\right)^2\right] = -e(d-2)\phi\left(v^\mu\partial_\mu\phi\right)\left(v^\nu\partial_\nu\omega\right)\,, \tag{3.7}$$

which precisely cancels against the transformation (3.6) of the conformal coupling. This means the timelike action (3.3) is indeed invariant under Weyl transformations.

## 3.2 Energy-momentum tensor

Following the general procedure described in Section 2.5, we now obtain the energy-momentum tensor for the timelike action (3.3) by varying with respect to the background Carroll geometry. Here we will focus on small variations of the flat space defined in Equation (2.6), in a boost frame where $\tau_\mu dx^\mu = -dt$. Varying the conformal coupling around this background, we get

$$\delta\left[K^{\mu\nu}K_{\mu\nu}+K^2-2v^\rho\partial_\rho K\right]\big|_{\text{flat}} = -2v^\rho\partial_\rho\,\delta K\big|_{\text{flat}} = -h_{\mu\nu}v^\rho v^\lambda\partial_\rho\partial_\lambda\delta h^{\mu\nu} + 2v^\mu h^\nu_\rho\partial_\mu\partial_\nu\delta v^\rho\,, \tag{3.8}$$

so that we obtain the following responses

$$T_\mu^\nu = \frac{1}{2}\tau_\mu\left(\nu^\rho\partial_\rho\phi\right)^2 - \frac{1}{2}\frac{d}{d-1}h_\mu^\rho\partial_\rho\phi\left(\nu^\lambda\partial_\lambda\phi\right) + \frac{1}{2}\frac{d-2}{d-1}h_\mu^\rho\nu^\lambda\phi\partial_\rho\partial_\lambda\phi\,,\tag{3.9a}$$

$$T_{\mu\nu}^h = \frac{1}{2}\frac{1}{d-1}h_{\mu\nu}\left(\nu^\rho\partial_\rho\phi\right)^2 - \frac{1}{2}\frac{d-2}{d-1}h_{\mu\nu}\phi\nu^\rho\nu^\lambda\partial_\rho\partial_\lambda\phi\,.\tag{3.9b}$$

The resulting energy-momentum tensor $T^\mu{}_\nu$ agrees with the $c \to 0$ limit of the energy-momentum tensor of the Lorentzian theory (3.1). Note that $T_\mu^\nu$ transforms under boosts, in agreement with the general identity (2.29). The components of the flat space boost-invariant energy-momentum tensor are then

$$T^0{}_0 = \frac{1}{2}\dot\phi^2\,,\tag{3.10a}$$

$$T^i{}_0 = 0\,,\tag{3.10b}$$

$$T^0{}_i = \frac{1}{2}\frac{d}{d-1}\dot\phi\partial_i\phi - \frac{1}{2}\frac{d-2}{d-1}\phi\partial_i\dot\phi\,,\tag{3.10c}$$

$$T^i{}_j = -\frac{1}{2}\frac{1}{d-1}\dot\phi^2\delta^i{}_j + \frac{1}{2}\frac{d-2}{d-1}\phi\ddot\phi\delta^i{}_j\,,\tag{3.10d}$$

where $\dot f = \nu^\mu\partial_\mu f$ denotes time derivatives. In contrast to the intermediate result (3.9), these quantities are invariant under local boost (even though they do transform under the global boost coordinate isometries of the flat background). Additionally, we see that $T^i{}_0$ vanishes, in accordance with the boost Ward identity (2.31). Furthermore, the trace of the energy-momentum tensor is

$$T^\mu{}_\mu = \frac{1}{2}(d-2)\phi\ddot\phi\,,\tag{3.11}$$

which vanishes on shell in general dimensions using the equation of motion $\ddot\phi = 0$. This energy-momentum tensor coincides in flat space with the Noether current corresponding to translational symmetry (see for example [22]), up to improvement terms. As in the relativistic case, there is an inherent ambiguity in the choice of the improvement terms and in the coupling of the action to a curved background. From the action with conformal coupling in Equation (3.3), by varying the Carroll background around flat space, we find an energy-momentum tensor that satisfies both the boost and Weyl Ward identities.

## 4 Spacelike action and dimensional reduction

In the preceding, we obtained a timelike conformal Carroll action by directly taking the $c \to 0$ limit of the rewriting (3.2) of the Lorentzian conformal scalar action. We can also take the limit in a different way, which will produce an action with spacelike derivatives in the kinetic term. This mirrors earlier work in the context of Carroll limits of electromagnetism, $p$-form gauge theories, higher-spin theories and gravity [21, 27, 28]. Here, our result will be a spacelike Carroll action for a conformally coupled scalar.

First, using the total derivative identity (2.21), we can rewrite the leading-order terms in the conformal coupling of the Lorentzian action (3.2) as follows,

$$\frac{(d-2)}{4(d-1)}\left[\mathcal{K}^{\mu\nu}\mathcal{K}_{\mu\nu} + \mathcal{K}^2 - 2V^\mu\partial_\mu\mathcal{K}\right]\phi^2 \approx \frac{(d-2)}{(d-1)}\phi\mathcal{K}V^\mu\partial_\mu\phi + \frac{(d-2)}{4(d-1)}\left[\mathcal{K}^{\mu\nu}\mathcal{K}_{\mu\nu} - \mathcal{K}^2\right]\phi^2\,.\tag{4.1}$$

Using this observation, we can rewrite the leading kinetic and conformal coupling terms in the

Lorentzian scalar action (3.2) as a sum of two squares,

$$-V^\mu V^\nu \partial_\mu \phi \partial_\nu \phi + \frac{(d-2)}{4(d-1)} \left[ \mathcal{K}^{\mu\nu}\mathcal{K}_{\mu\nu} + \mathcal{K}^2 - 2V^\mu \partial_\mu \mathcal{K} \right] \phi^2 \tag{4.2}$$

$$\approx -\left( V^\mu \partial_\mu \phi - \frac{(d-2)}{2(d-1)}\phi \mathcal{K} \right)^2 + \frac{(d-2)}{4(d-1)} \left[ \mathcal{K}^{\mu\nu}\mathcal{K}_{\mu\nu} - \frac{1}{d-1}\mathcal{K}^2 \right] \phi^2 \tag{4.3}$$

$$= -\left( V^\mu \partial_\mu \phi - \frac{(d-2)}{2(d-1)}\phi \mathcal{K} \right)^2 + \frac{(d-2)}{4(d-1)} \bar{G}^{\mu\nu\rho\sigma}\mathcal{K}_{\mu\nu}\mathcal{K}_{\rho\sigma}\phi^2 . \tag{4.4}$$

Here, we have introduced the tensor $\bar{G}^{\mu\nu\rho\sigma}$ which, together with its inverse, is given by

$$\bar{G}^{\mu\nu\rho\sigma} = \frac{1}{2} \left( \Pi^{\mu\rho}\Pi^{\nu\sigma} + \Pi^{\mu\sigma}\Pi^{\nu\rho} - \frac{2}{d-1}\Pi^{\mu\nu}\Pi^{\rho\sigma} \right) , \tag{4.5}$$

$$\bar{G}_{\mu\nu\rho\sigma} = \frac{1}{2} \left( \Pi_{\mu\rho}\Pi_{\nu\sigma} + \Pi_{\mu\sigma}\Pi_{\nu\rho} - \frac{2}{d}\Pi_{\mu\nu}\Pi_{\rho\sigma} \right) . \tag{4.6}$$

Note that these satisfy $\bar{G}^{\mu\nu\rho\sigma}\bar{G}_{\rho\sigma\gamma\delta} = \Pi_{\gamma\rho}\Pi_{\delta\sigma}\Pi^{\rho(\mu}\Pi^{\nu)\sigma} - \Pi^{\mu\nu}\Pi_{\gamma\delta}/(d-1)$, which is the identity operator on the space of symmetric spatial traceless two-tensors. As a result, $\bar{G}^{\mu\nu\rho\sigma}$ can be regarded as a modified version of the deWitt metric, which can be unambiguously inverted on the space of symmetric spatial traceless two-tensors. This mirrors the construction of the covariant spacelike (or magnetic) Carroll gravity theory in [28], where the regular deWitt metric enters.

With this, we can then rewrite the leading-order terms in the action (3.2) using a scalar Lagrange multiplier $\chi$ and a symmetric traceless Lagrange multiplier $X_{\mu\nu}$

$$\frac{c^2}{4}\chi^2 + \chi \left( V^\mu \partial_\mu \phi - \frac{(d-2)}{2(d-1)}\phi \mathcal{K} \right) - \frac{c^2}{4}\bar{G}^{\mu\nu\rho\sigma}X_{\mu\nu}X_{\rho\sigma} + \bar{G}^{\mu\nu\rho\sigma}X_{\mu\nu}\mathcal{K}_{\rho\sigma}\phi . \tag{4.7}$$

Integrating out $X_{\mu\nu}$ and $\chi$ reproduces (4.4) times an overall factor of $1/c^2$, which is how these terms appear inside the action (3.2). However, by writing these terms as in (4.7), we have lowered their overall degree in $c^2$, so that the $c \to 0$ limit of the Lorentzian conformal scalar action (3.2) now leads to an inequivalent result. Rescaling the fields $\phi$, $\chi$ and $\chi_{\mu\nu}$ with a factor of $c^{-1}$ for the overall scaling, we obtain

$$S = -\frac{1}{2} \int d^d x \, e \left[ h^{\mu\nu}\partial_\mu \phi \partial_\nu \phi + \frac{(d-2)}{4(d-1)} \left( h^{\mu\nu}\tilde{R}_{\mu\nu} - \tilde{\nabla}_\mu a^\mu \right) \phi^2 + \chi \left( v^\mu \partial_\mu \phi - \frac{(d-2)}{2(d-1)}K\phi \right) + \chi^{\mu\nu}K_{\mu\nu}\phi \right] . \tag{4.8}$$

Here, $a^\mu = h^{\mu\nu}a_\nu$ is the dual vector to the spatial acceleration one-from (2.8) and $\chi^{\mu\nu} = G^{\mu\nu\rho\sigma}X_{\rho\sigma}$ is a symmetric spatial traceless tensor.

In this action, spatial derivatives dominate, so it is clearly distinct from the timelike action (3.3). In addition, the timelike action identically vanishes when the constraints imposed by the Lagrange multipliers are applied. Thus, we conclude that we cannot consider both actions at the same time, in contrast to [24, 25]. As we will check explicitly below, the result (4.8) is also invariant under both Weyl transformations and local Carroll boosts, so it describes a different, 'spacelike' notion of conformal Carroll scalars. Furthermore, we will derive its energy-momentum tensor, and we will verify that it satisfies the classical Ward identities that were established in Section 2.5. Finally, we show that it is possible to effectively reduce the $d$-dimensional conformal Carroll scalar action (4.8) to a $(d-1)$-dimensional action for a Euclidean conformally coupled particle.

A curved space Carroll action similar to (4.8) has also been derived in [24], and it was recently reconsidered in [25]. However, this did not include the constraint terms. As we will see below, these constraints are essential to maintaining local Carroll boost invariance. This is to be expected, since we argued in Section 2.7 that the presence of local Carroll boosts is the crucial difference between our notion of Carroll geometry and the formalism employed in [24–26].

## 4.1 Carroll boost and Weyl symmetries

We now show explicitly that the spacelike action (4.8) is invariant under boosts and Weyl transformations. For this, we can either retain the Lagrange multipliers, or we can integrate them out and work on the constraint surface. We will mainly take the latter route, but we also list the transformations of the Lagrange multipliers that result in the invariance of the action away from the constraint surface.

The constraints imposed by the Lagrange multipliers are

$$v^\mu \partial_\mu \phi - \frac{(d-2)}{2(d-1)} \phi K = 0 , \tag{4.9a}$$

$$K_{\mu\nu} - \frac{K}{d-1} h_{\mu\nu} = 0 . \tag{4.9b}$$

Note that the Lagrange multiplier $\chi^{\mu\nu}$ is traceless, which is why the trace of $K_{\mu\nu}$ is unconstrained. Since they only contain boost-invariant tensors, the constraints are invariant under local Carroll boosts. Likewise, using the transformations (2.24) of the extrinsic curvature under Weyl transformations, one can check that the constraints transform homogeneously with weight $-d/2$ and 1, respectively.

**Weyl symmetry.** First, we show that the first two terms in the action (4.8) are Weyl invariant on the constraint surface. From Equation (2.26), we can derive that the curvature terms transform as

$$\delta_\omega (h^{\mu\nu} \tilde{R}_{\mu\nu} - \tilde{\nabla}_\mu a^\mu) = -2\omega(h^{\mu\nu} \tilde{R}_{\mu\nu} - \tilde{\nabla}_\mu a^\mu) + 2(1-d)\tilde{\nabla}_\mu(h^{\mu\nu} \tilde{\nabla}_\nu \omega) . \tag{4.10}$$

On the other hand, the kinetic term transforms as

$$\begin{aligned} \delta_\omega(h^{\mu\nu} \partial_\mu \phi \partial_\nu \phi) &= -d h^{\mu\nu} \partial_\mu \phi \partial_\nu \phi + (2-d) h^{\mu\nu} \phi \partial_\mu \phi \partial_\nu \omega \\ &\approx -d h^{\mu\nu} \partial_\mu \phi \partial_\nu \phi - \frac{2-d}{2} \tilde{\nabla}_\mu(h^{\mu\nu} \tilde{\nabla}_\nu \omega) \phi^2 , \end{aligned} \tag{4.11}$$

where we integrated by parts in the last equality. Adding (4.10) and (4.11) using their relative coefficients in the action (4.8), we see that the inhomogeneous terms cancel, leaving the action invariant under Weyl transformations.

Alternatively, we can keep the Lagrange multipliers $\chi$ and $\chi^{\mu\nu}$ in the action and specify their Weyl transformations. As the kinetic and the conformal coupling are invariant by themselves, the transformation laws for the Lagrange multipliers are homogeneous and their task is to balance the Weyl weight of the constraints, so that we can take

$$\delta_\omega \chi = -\frac{d}{2} \omega \chi , \qquad \delta_\omega \chi^{\mu\nu} = -\frac{d+4}{2} \omega \chi^{\mu\nu} . \tag{4.12}$$

With these transformations, the action (4.8) is also Weyl invariant without imposing the constraints.

**Boost symmetry.** We now turn to demonstrating the boost invariance of the spacelike action, again first by assuming that the constraint equations hold. For the kinetic term of the spacelike action we find

$$\begin{aligned} \delta_\lambda(h^{\mu\nu} \partial_\mu \phi \partial_\nu \phi) &= 2v^\mu \lambda^\nu \partial_\mu \phi \partial_\nu \phi = \frac{d-2}{2(d-1)} K \lambda^\mu \partial_\mu \phi^2 \\ &\approx -\frac{d-2}{2(d-1)} \tilde{\nabla}_\mu(K\lambda^\mu) \phi^2 , \end{aligned} \tag{4.13}$$

where we used the constraint (4.9a) in the second equation. On the other hand, the boost variation of the curvature terms appearing in the action is

$$
\begin{aligned}
\delta_\lambda(h^{\mu\nu}\tilde{R}_{\mu\nu} - \tilde{\nabla}_\mu a^\mu)\phi^2 &= 2\phi^2\tilde{\nabla}_\rho(\lambda^\rho K - \nu^\rho\tilde{\nabla}_\mu\lambda^\mu) + 2\phi^2 a^\mu\lambda^\nu K_{\mu\nu} + 2\phi^2 K^{\mu\nu}\tilde{\nabla}_\mu\lambda_\nu \\
&= 2\phi^2\tilde{\nabla}_\rho(\lambda^\rho K - \nu^\rho\tilde{\nabla}_\mu\lambda^\mu) + \frac{2K\phi^2}{d-1}(h^{\mu\nu}\tilde{\nabla}_\mu\lambda_\nu + a^\mu\lambda_\nu) \\
&\approx 2\tilde{\nabla}_\mu(K\lambda^\mu)\phi^2,
\end{aligned}
\tag{4.14}
$$

where we used a variational identity for the Ricci tensor in the first equality. In the second equality we imposed the constraint (4.9b) and in the last equality we integrated by parts before using the constraint (4.9a). Plugging Equations (4.13) and (4.14) into the action (4.8), we conclude that it is indeed boost-invariant on the constraint surface.

Alternatively, we can retain the Lagrange multipliers and impose

$$
\delta_\lambda\chi = -2\lambda^\nu\partial_\nu\phi - \frac{d-2}{d-1}\phi\tilde{\nabla}_\mu\lambda^\mu,
\tag{4.15}
$$

$$
\delta_\lambda\chi^{\mu\nu} = -\frac{d-2}{2(d-1)}\phi\left(a^{(\mu}\lambda^{\nu)} - \frac{a_\rho\lambda^\rho}{d-1}h^{\mu\nu} + h^{\rho(\mu}\tilde{\nabla}_\rho\lambda^{\nu)} - \frac{\tilde{\nabla}_\rho\lambda^\rho}{d-1}h^{\mu\nu}\right).
\tag{4.16}
$$

These boost transformations of the Lagrange multipliers precisely cancel the terms in the other transformations that previously vanished due to their associated constraints. Either way, the spacelike action (4.8) is invariant under local Carroll boosts.

## 4.2 Energy-momentum tensor

Following our general procedure in Section 2.5, the energy-momentum tensor can be computed by varying the spacelike action (4.8) with respect to the background Carroll geometry. Here, we will focus on fluctuations around the flat background (2.6), in a boost frame where $\tau_\mu dx^\mu = -dt$. First, from the definitions it is straight forward that

$$
h^{\nu\sigma}\delta\tilde{R}_{\nu\sigma}\big|_{\text{flat}} = h^{\rho\gamma}h^{\alpha\beta}\left(\partial_\rho\partial_\alpha\delta h_{\gamma\beta} - \partial_\rho\partial_\gamma\delta h_{\alpha\beta}\right)\big|_{\text{flat}},
\tag{4.17}
$$

$$
\delta\tilde{\nabla}_\mu a^\mu\big|_{\text{flat}} = h^{\mu\epsilon}\nu^\delta\partial_\mu\partial_\delta\delta\tau_\epsilon + h^{\mu\epsilon}\tau_\delta\partial_\mu\partial_\epsilon\delta\nu^\delta\big|_{\text{flat}},
\tag{4.18}
$$

which leads to the following variation of the conformal coupling term,

$$
\begin{aligned}
&\delta\left(h^{\mu\nu}\tilde{R}_{\mu\nu} - \tilde{\nabla}_\mu a^\mu\right)\big|_{\text{flat}} \\
&= -2\tau_{,\gamma}h^{\alpha\beta}\partial_\alpha\partial_\beta\delta_\eta\nu^\nu + \left[-h^\rho{}_\mu h^\alpha{}_\nu\partial_\rho\partial_\alpha + h_{\mu\nu}h^{\rho\gamma}\partial_\rho\partial_\gamma + \nu^\rho\partial_\rho\tau_{(\mu}\partial_{\nu)}\right]\delta h^{\mu\nu}\big|_{\text{flat}},
\end{aligned}
\tag{4.19}
$$

where we used $h^{\mu\nu}\delta\tau_\nu = -\tau_{,\nu}\delta h^{\mu\nu}$. In addition, it is convenient to use

$$
\delta(\chi^{\mu\nu}K_{\mu\nu}) = \frac{1}{2}\chi_{\mu\nu}\nu^\rho\partial_\rho\delta h^{\mu\nu} - \chi^{\rho\sigma}h_{\mu\sigma}\partial_\rho\delta_\eta\nu^\mu,
\tag{4.20}
$$

where $\bar{G}^{\rho\sigma\alpha\beta}$ is a projector as defined in (4.5) and we allow for raising and lowering the indices of $\chi^{\mu\nu}$ using the spatial metric.

For compactness, we introduce the notation $\hat{\partial}_\mu = h_\mu^\nu\partial_\nu$, where $h_\mu^\nu$ is the spatial projector defined below Equation (2.1). Using the identities (4.19) and (4.20), the constraint (4.9a) which in flat space implies $\nu^\mu\partial_\mu\phi = 0$, and the equation of motion $\hat{\partial}^2\phi = -\frac{1}{2}\nu^\rho\partial_\rho\chi$, we find that the on-shell energy-momentum tensor is given by

$$
T^\nu_\nu = \frac{-1}{d-1}\tau_{,\nu}(\hat{\partial}\phi)^2 - \frac{1}{2}\frac{d-2}{d-1}\tau_{,\nu}\phi\nu^\rho\partial_\rho\chi - \chi\hat{\partial}_\nu\phi + \frac{1}{2}\frac{d-2}{d-1}\hat{\partial}_\nu(\chi\phi) - \hat{\partial}_\alpha(\chi^\alpha{}_\nu\phi),
\tag{4.21a}
$$

$$
T^h_{\mu\nu} = \frac{1}{d-1}(\hat{\partial}\phi)^2 h_{\mu\nu} - \frac{d}{d-1}\hat{\partial}_\mu\phi\hat{\partial}_\nu\phi + \frac{d-2}{d-1}\phi\hat{\partial}_\mu\hat{\partial}_\nu\phi + \nu^\rho\partial_\rho(\chi_{\mu\nu}\phi).
\tag{4.21b}
$$

In terms of the flat space coordinates $x^\mu = (t, x^i)$, the resulting components of the boost-invariant energy-momentum tensor (2.30) are given by

$$T^0{}_0 = -\frac{1}{d-1}(\partial_i \phi \partial^i \phi) - \frac{1}{2}\frac{d-2}{d-1}\phi v^\rho \partial_\rho \chi, \tag{4.22a}$$

$$T^i{}_0 = 0, \tag{4.22b}$$

$$T^0{}_i = \chi \partial_i \phi - \frac{1}{2}\frac{d-2}{d-1}\partial_i(\chi \phi) + \partial_j(\chi^j{}_i \phi), \tag{4.22c}$$

$$T^i{}_j = -\frac{1}{d-1}(\partial_i \phi \partial^i \phi)\delta^i{}_j + \frac{d}{d-1}\partial^i \phi \partial_j \phi - \frac{d-2}{d-1}\phi \partial^i \partial_j \phi - v^\rho \partial_\rho(\chi^i{}_j \phi). \tag{4.22d}$$

We note that by putting $\chi_{ij} = 0$, and upon identifying $\chi$ as canonical momentum, we find exactly the same momentum structure $T^0{}_i$ as in the timelike case, where $\dot\phi$ plays the role of canonical momentum. Using the fact that the spatial symmetric tensor $\chi_{ij}$ is traceless, we find

$$T^0{}_0 + T^i{}_i = -\frac{d-2}{d-1}\left(\frac{1}{2}\phi v^\rho \partial_\rho \chi + \phi \partial_i \partial^i \phi\right) = 0, \tag{4.23}$$

which is zero due to the equations of motion. This shows that the classical energy-momentum tensor is traceless for generic dimensions, as a consequence of the Weyl invariance of the action (4.8) and upon using the equations of motion.

## 4.3  Reduction to lower-dimensional Euclidean CFT

On the flat Carroll background (2.6), the spacelike action (4.8) takes the form

$$S = \frac{1}{2}\int d^d x\, e\left[-\chi \dot\phi - \delta^{ij}\partial_i \phi \partial_j \phi\right]. \tag{4.24}$$

Note that the transformations of $h^{\mu\nu}\partial_\mu\partial_\nu = \delta^{ij}\partial_i\partial_j$ under boosts are precisely cancelled by the constraint that $v^\mu\partial_\mu\phi = \dot\phi = 0$, so the action above is indeed boost-invariant. From a Hamiltonian perspective, we have $\pi_\phi = -\chi$ and $\mathcal{H} = \delta^{ij}\partial_i\phi\partial_j\phi$. The result is that the field $\phi$ has no time evolution, only a non-trivial spatial profile along the $x^i$ directions is possible. On this background, the conformal Carroll scalar therefore effectively reduces to a scalar particle coupled to a flat Euclidean background.

With this in mind, it seems reasonable to ask if the spacelike action (4.8) can also be dimensionally reduced for a curved Carroll background. We would then expect the result to be a Euclidean conformal field theory on a general $(d-1)$-dimensional Riemannian geometry with $h_{\mu\nu}$ as its metric. After all, the Carrollian time evolution $v^\mu\partial_\mu\phi$ of the scalar field is fixed by the constraint (4.9a). We now show that this is indeed the case.

**Integrating out the constraints**   To be precise, we will show in this section that the action (4.8) for a conformally coupled spacelike Carroll scalar in $d$ spacetime dimensions can be reduced to the Euclidean version of the usual conformal scalar action (3.1) on a $(d-1)$-dimensional spatial slice, together with an additional background field. This reduction is possible due to the constraints (4.9) on the configuration space and the background geometry, which are simple enough to allow for explicit integration.

The constraints in the spacelike action fix the evolution of $\phi$ and $h_{\mu\nu}$ along $v^\mu$. It is therefore useful to introduce adapted coordinates $x^\mu = (t, x^i)$ such that

$$v^\mu\partial_\mu = \alpha^{-1}\partial_t, \qquad h_{\mu\nu}dx^\mu dx^\nu = h_{ij}dx^i dx^j, \qquad \tau_\mu dx^\mu = -\alpha dt + b_i dx^i. \tag{4.25}$$

Here we have introduced a lapse function $\alpha(t, x^i)$, which will be useful later on, and $i, j, \ldots = 1, \ldots, d-1$ are spatial indices. As we discussed around Equation (2.5), the clock one-form $\tau_\mu$ is not invariant under Carroll boosts $\lambda_i$, which act by shifting $b_i$. We can therefore set $b_i = 0$ using boosts, so that

$$v^\mu \partial_\mu = \alpha^{-1} \partial_t, \qquad h_{\mu\nu} dx^\mu dx^\nu = h_{ij} dx^i dx^j, \qquad \tau_\mu dx^\mu = -\alpha dt. \tag{4.26}$$

The resulting $\tau_\mu$ defines a spatial foliation consisting of constant $t$ slices, and we will first perform the reduction of the spacelike action in this boost frame. We will argue afterwards that a generic spatial foliation defined by a $\tau_\mu$ with $b_i \neq 0$ can be included using a coordinate transformation that acts as a Weyl transformation in the lower-dimensional action.

Recalling that $K_{\mu\nu} = -\frac{1}{2} \mathcal{L}_v h_{\mu\nu}$, we see that the constraints (4.9) written in terms of the coordinates (4.26) imply

$$\dot{\phi}(t, x^i) = \frac{d-2}{2(d-1)} \alpha(t, x^i) K(t, x^i) \phi(t, x^i), \tag{4.27a}$$

$$\dot{h}_{ij}(t, x^i) = -\frac{2}{d-1} \alpha(t, x^i) K(t, x^i) h_{ij}(t, x^i), \tag{4.27b}$$

where the dot denotes the $t$ derivative. Additionally, it is useful to introduce the determinant $h = \det h_{ij}$ of the spatial metric. Its time evolution follows from

$$\mathcal{L}_v e = -e v^\mu a_\mu - e h^{\mu\nu} K_{\mu\nu} = -eK = v^\rho \partial_\rho e + e \partial_\rho v^\rho, \tag{4.28}$$

where we have used (2.2) in the first equality. Since $e = \alpha \sqrt{h}$, this implies

$$\dot{h}(t, x) = -2\alpha(t, x^i) K(t, x^i) h(t, x^i). \tag{4.29}$$

We can integrate these first-order ordinary differential equations (4.27) and (4.29) for a given background. Since these differential equations are all of the same form, the resulting time evolutions are related. In particular, it is convenient to write the time evolution of $\phi(t, x^i)$ and $h_{ij}(t, x^i)$ in terms of the time evolution of $h(t, x^i)$,

$$\phi(t, x^i) = \phi(0, x^i) \left( \frac{h(t, x^i)}{h(0, x^i)} \right)^{\frac{2-d}{4(d-1)}}, \qquad h_{ij}(t, x^i) = h_{ij}(0, x^i) \left( \frac{h(t, x^i)}{h(0, x^i)} \right)^{\frac{1}{d-1}}. \tag{4.30}$$

As the time dependence of these fields is a purely multiplicative conformal factor, we can scale it out using a finite Weyl transformation without changing the spacelike action (4.8), since the action is Weyl-invariant. To absorb the time dependence of these quantities, we can act with the following Weyl transformation,

$$h_{ij}(t, x^i) \to \left( \frac{h(t, x^i)}{h(0, x^i)} \right)^{-\frac{1}{d-1}} h_{ij}(t, x^i) = h_{ij}(0, x), \tag{4.31a}$$

$$\phi(t, x^i) \to \left( \frac{h(t, x^i)}{h(0, x)} \right)^{-\frac{2-d}{4(d-1)}} \phi(t, x^i) = \phi(0, x^i). \tag{4.31b}$$

Note that the powers in (4.30) precisely allow us to cancel the time dependence of both objects at the same time. The Weyl transformation also acts on $\tau_\mu$, and using the parametrization (4.26) this means it acts on $\alpha(t, x)$ as

$$\alpha(t, x^i) \to \alpha(t, x^i) \left( \frac{h(t, x^i)}{h(0, x^i)} \right)^{-\frac{1}{2(d-1)}} = \alpha'(t, x^i). \tag{4.32}$$

After this Weyl transformation, we have therefore moved all the time dependence into the (modified) lapse function $\alpha'(t, x^i)$. This only fixes the time-dependent part of the Weyl symmetry of the action, and $x^i$-dependent Weyl transformations are still allowed.

**Rewriting to Riemannian quantities** Furthermore, to relate the Carrollian objects to the Riemannian geometry induced on the constant $t$ slices, we need to define the spatially projected derivative

$$\hat{\nabla}_\rho X_{\mu\nu} = h_\rho^\gamma h_\mu^\alpha h_\nu^\beta \tilde{\nabla}_\gamma X_{\alpha\beta}, \tag{4.33}$$

which can be shown to coincide with the Levi-Civita connection induced by the pull-back of $h_{\mu\nu}$. Having defined a covariant derivative on the spacelike hypersurface, following the discussion in [28], one can derive a Gauss-type equation relating the Ricci of the $\tilde{\nabla}$ and $\hat{\nabla}$ connections

$$\hat{R} = h^{\mu\nu}\tilde{R}_{\mu\nu} + \hat{\nabla}_\mu a^\mu + a_\mu a^\mu. \tag{4.34}$$

Along with the coordinate-specific identities $e = \alpha\sqrt{h}$ and $\hat{\nabla}_\mu a^\mu + a_\mu a^\mu = \alpha^{-1}\hat{\nabla}^2\alpha$, the Gauss equation allows us to rewrite the action (4.8) on the constraint surface as

$$S = -\frac{1}{2}\int d^d x \, e\left[h^{ij}\partial_i\phi\partial_j\phi + \frac{(d-2)}{4(d-1)}(\hat{R} - 2(\hat{\nabla}_\mu a^\mu + a_\mu a^\mu))\phi^2\right] \tag{4.35}$$

$$= -\frac{1}{2}\int d^{d-1}x\sqrt{h}\left(\int dt\,\alpha'\right)\left[h^{ij}\partial_i\phi\partial_j\phi + \frac{(d-2)}{4(d-1)}\hat{R}\right] - \frac{1}{2}\int d^{d-1}x\sqrt{h}\frac{-2(d-2)}{4(d-1)}\phi^2\hat{\nabla}^2\left(\int dt\,\alpha'\right),$$

where we used the fact that only $\alpha'$ has a $t$-dependence to move the $t$-integral. The form of the action (4.35) suggests defining the quantity

$$A(x^i) = \int dt\,\alpha'(t,x^i) = \int dt\,\alpha(t,x^i)\left(\frac{h(t,x^i)}{h(0,x^i)}\right)^{-\frac{1}{2(d-1)}}. \tag{4.36}$$

This transforms as a scalar from the point of view of the spatial $x^i$ diffeomorphisms. Under the remaining spatial Weyl transformations, $A$ inherits the transformation of $\alpha$,

$$\delta_\omega A = \omega A. \tag{4.37}$$

Inserting the definition of $A$ back into the action (4.35), we can rewrite it as

$$S = -\frac{1}{2}\int d^{d-1}x\,\sqrt{h}\left[h^{ij}\partial_i\hat{\phi}\partial_j\hat{\phi} + \frac{(d-3)}{4(d-2)}\hat{R}\hat{\phi}^2\right. \tag{4.38}$$

$$\left. + \frac{1}{4(d-1)(d-2)}(\hat{R} - (d-1)(d-2)A^{-2}h^{ij}\partial_i A\partial_j A + 2(d-2)A^{-1}\hat{\nabla}^2 A)\hat{\phi}^2\right].$$

In the first line, we collected the terms which make up the action for a Euclidean relativistic CFT in $d-1$ dimensions, with $h_{ij}$ as the Riemannian metric. Furthermore, we defined the $(d-1)$-dimensional scalar field as

$$\hat{\phi} = A^{1/2}\phi \quad \Rightarrow \quad \delta_\omega\hat{\phi} = \frac{3-d}{2}\omega\hat{\phi}, \tag{4.39}$$

in anticipation of $\hat{\phi}$ being a conformal scalar in $(d-1)$ dimensions, *i.e.*, it has the correct Weyl weight. Remarkably, we can recognize the last term in (4.38) as a finite Weyl rescaling of a Ricci scalar leading to the expression

$$S = -\frac{1}{2}\int d^{d-1}x\,\sqrt{h}\left[h^{ij}\partial_i\hat{\phi}\partial_j\hat{\phi} + \frac{(d-3)}{4(d-2)}\hat{R}\hat{\phi}^2 + \frac{1}{4(d-1)(d-2)}A^{-2}\hat{R}_{A^{-2}h_{ij}}\hat{\phi}^2\right], \tag{4.40}$$

where $\hat{R}_{A^{-2}h_{ij}}$ denotes the Ricci scalar calculated with respect to the metric $A^{-2}h_{ij}$. The final form of the reduced action (4.40) makes the Euclidean Weyl symmetry manifest: the first two

terms constitute the usual relativistic scalar action, while the non-trivial reminder $\hat{R}_{A^{-2}h_{ij}}$ is inert since $A^{-2}$ and $h_{ij}$ transforms oppositely.

Finally, since we performed the reduction in a particular boost frame (4.26), we should investigate what happens in a general boost frame (4.25) where $b_i \neq 0$. Assuming the corresponding clock one-form $\tau_\mu$ satisfies the Frobenius condition $\tau \wedge d\tau = 0$, since we would otherwise not have a spatial foliation on which we can reduce, we can find a new adapted set of coordinates $\tilde{x}^\mu = (\tilde{t}, x^i) = (\tilde{t}(t, x^i), x^i)$ such that

$$v^\mu \partial_\mu = \tilde{\alpha}^{-1} \partial_{\tilde{t}}, \qquad h_{\mu\nu} dx^\mu dx^\nu = h_{ij} dx^i dx^j, \qquad \tau_\mu dx^\mu = -\tilde{\alpha} d\tilde{t}. \qquad (4.41)$$

However, the spatial slice $\tilde{t} = 0$ does not necessarily correspond to the $t = 0$ slice. On the other hand, since we do not modify the $x^i$ coordinates, these slices are just related by a (possibly $x^i$-dependent) evolution in time. We therefore have

$$\phi(t(\tilde{t} = 0, x^i), x^i) = \tilde{\Omega}(x^i)^{-\frac{d-2}{2}} \phi(t = 0, x^i), \qquad (4.42a)$$

$$h_{ij}(t(\tilde{t} = 0, x^i), x^i) = \tilde{\Omega}(x^i)^2 h_{ij}(t = 0, x^i), \qquad (4.42b)$$

$$\tilde{\alpha}'(\tilde{t}, x^i) = \frac{\partial t}{\partial \tilde{t}} \tilde{\Omega}(x^i) \alpha'(t(\tilde{t}, x^i), x^i). \qquad (4.42c)$$

The first two lines follow from (4.30) and the last line can be obtained using the coordinate transformation $\tilde{t}(t, x^i)$ in either $v^\mu \partial_\mu$ or $\tau_\mu dx^\mu$ together with the definition (4.32). Integrating the last line over $\tilde{t}$, we obtain

$$\tilde{A}(x^i) = \int d\tilde{t}\, \tilde{\alpha}'(\tilde{t}, x^i) = \tilde{\Omega}(x^i) A(x^i). \qquad (4.43)$$

Consequently, we see that all fields entering in the reduced action (4.40) transform under a spatial Weyl transformation $\tilde{\Omega}(x^i)$, which is given by

$$\tilde{\Omega}(x^i) = \left( \frac{h(t(\tilde{t} = 0, x^i), x^i)}{h(t = 0, x^i)} \right). \qquad (4.44)$$

Since the reduced action is precisely invariant under spatial Weyl transformations, we conclude that any boost frame corresponding to a spatial foliation of our $d$-dimensional Carroll spacetime leads to the same $(d-1)$-dimensional Euclidean conformal action (4.40).

**Relation to embedding space construction?** With these results in mind, it is tempting to draw an analogy to the embedding space construction that is frequently used in relativistic conformal field theory, see for example the lectures [62]. This construction relies on the fact that the Euclidean conformal group $SO(d, 1)$ in $(d-1)$ dimensions is equivalent to the Lorentz group of isometries of $\mathbb{R}^{d,1}$. Since the latter is linearly realized on the higher-dimensional fields, this is a useful tool to construct spinning representations of the conformal algebra.

A crucial step in this construction is the reduction of the $(d + 1)$-dimensional Lorentzian degrees of freedom to the Euclidean $\text{CFT}_{d-1}$. For this, the Euclidean submanifold can be recovered as the projective identification of the light cone at the origin of $\mathbb{R}^{d,1}$ along null rays. This is reminiscent of our dimensional reduction of the spacelike action. The light cone in $\mathbb{R}^{d,1}$ carries a natural Carroll structure, as illustrated in Figure 2, where the null direction along the light cone corresponds to the Carroll time direction. The projective identification along the null rays then appears to be similar to our process of integrating out the Carroll time. Remarkably, our reduction of a spatial conformal Carroll theory to a $\text{CFT}_{d-1}$ takes place on the level of the action, whereas (to our knowledge) the embedding space construction has only been used to construct representations. It would be very interesting to make this analogy more precise.

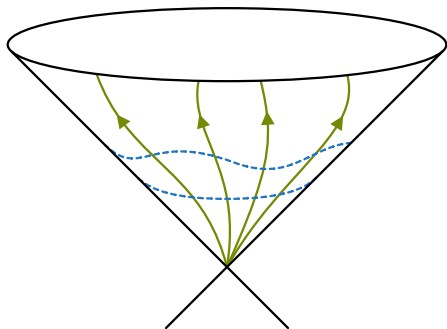

Figure 2: Carroll structure on the light cone of the embedding space formalism.

## 5 Discussion

We derived two distinct scalar field actions that are invariant under local Carroll boosts, Weyl transformations and general diffeomorphisms on curved backgrounds. Corresponding to the derivatives that appear in their kinetic terms, we refer to these actions as *timelike* and *spacelike*, respectively. For both the timelike and spacelike actions, we derived a traceless energy-momentum tensor using the coupling to general curved Carroll geometries, and we showed that their energy fluxes vanish. This fact is a consequence of the invariance under local Carroll boosts and it is one of the main novelties of the models proposed in this work. In the spacelike case, our action contains a set of constraints that are essential for the invariance under local Carroll boosts. Additionally, we showed that the spacelike action can be dimensionally reduced to a Euclidean CFT on the spatial directions. In addition to the spatial metric, this CFT couples to a scalar that encodes its higher-dimensional pedigree.

A fundamental property of the actions that we derived in this work is their classical Weyl invariance. The breaking of this symmetry at quantum level is an important area of research [63] and has previously been considered in several non-Lorentzian theories [53–55, 64–66]. In the context of flat holography, there have been recent discussions on loop corrections to the celestial energy-momentum tensor [67–69] and the trace anomaly close to a flat background for two-dimensional BMS field theories [70]. We plan to make contact with both of these directions by performing the systematic classification of the trace anomaly for Carroll-invariant theories, using a cohomological approach. Furthermore, the conformal scalar actions presented in this work open the possibility for the computation of the trace anomaly in specific models, for example using heat kernel techniques [71, 72] (see [73–78] for applications in the non-relativistic case) or perturbative methods [79] (see [80] for the Lifshitz case). The computation of the conformal anomaly in explicit examples is not only a consistency check of the cohomology result, but will also determine the central charges of the models under consideration. Since trace anomalies can be used to characterize monotonicity quantities along the RG flow [81–83], these investigations could lead us to unravel further universal aspects of field theories with Carroll symmetry.

It is well known that the energy-momentum tensor of the relativistic conformal scalar can be cast in perfect fluid form. In order to investigate whether we can do the same for the timelike and spacelike scenarios we studied here, we can use the tensorial form of a general boost-agnostic perfect fluid energy-momentum tensor [31]

$$T^0{}_0 = -\mathcal{E}, \quad T^i{}_0 = -(\mathcal{E} + P)v^i, \quad T^0{}_j = \mathcal{P}_j, \quad T^i{}_j = P\delta^i{}_j + v^i\mathcal{P}_j, \tag{5.1}$$

where $\mathcal{E}$, $P$, $v^i$ and $\mathcal{P}_j$ respectively denote the energy density, pressure, fluid velocity and

momentum flux density. For the timelike case, we can identify $v^i = 0$ and

$$\mathcal{E} = -\frac{1}{2}\dot{\phi}^2, \qquad P = -\frac{1}{2}\frac{1}{d-1}\dot{\phi}^2 + \frac{1}{2}\frac{d-2}{d-1}\phi\ddot{\phi},$$
$$\mathcal{P}_i = -\frac{1}{2}\frac{d}{d-1}\dot{\phi}\partial_i\phi + \frac{1}{2}\frac{d-2}{d-1}\phi\partial_i\dot{\phi}.$$

(5.2)

The fact that this fluid is not moving is in line with general Carroll expectation. A special feature of this fluid is that it has non-vanishing momentum flux, despite having vanishing fluid velocity. Remarkably, this means we find an explicit example of a theory that defies the usual assumption that $\mathcal{P}_i \propto v^i$ and requires for $\mathcal{P}_i$ to be treated as a fundamental fluid variable, which was proposed as a possibility in [22]. This property would likely contribute to for example novel transport contributions [84–87] and would change the analysis of its stability [88]. On the other hand, at first sight it seems challenging to interpret the spacelike case as a perfect fluid.

Finally, the details of the quantization of these Carroll theories remain unclear. In the timelike theory, the absence of spatial derivatives means that the spatial coordinates appear only as a label for independent copies of the scalar field at each point. From this perspective, the timelike theory describes a collection of free particles with arbitrary energies. One could also consider adding a mass regulator [18], which breaks conformality. This leads to the usual harmonic oscillator spectrum at every independent spatial point instead, which is not continuously connected to the free particle spectrum.[5] Fully understanding the quantization of these conformal Carroll theories will surely require us to go further *down the rabbit hole*.

# Acknowledgements

We are thankful to Ronnie Rodgers for useful discussions, and to Luca Ciambelli, Francesco Alessio and Niels Obers for useful discussions and helpful comments on an early version of this draft. SB is supported by the Israel Science Foundation (grant No. 1417/21), the German Research Foundation through a German-Israeli Project Cooperation (DIP) grant "Holography and the Swampland", the Azrieli Foundation and the Kreitmann School of Advanced Graduate Studies. GO is supported by the Vetenskapsrådet project grant "Emergent Spacetime from Nonrelativistic Holography". Nordita is supported in part by NordForsk. WS is supported by the Icelandic Research Fund (IRF) via the Grant of Excellence titled "Quantum Fields and Quantum Geometry".

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
