# Peer review of "Conformal Carroll Scalars with Boosts"

_SciPost Physics, doi:SciPost Phys. 14, 086 (2023)_

## Round 2 · Referee Report · Anonymous (Referee 1) · 2022-10-28

Report

The manuscript presents a construction of two actions for scalars which exhibit invariance under local Carroll boosts and Weyl transformations: - a "timelike" action where time derivatives dominate; see eq. (3.3) - a "spacelike" action where spatial derivatives dominate; see eq. (4.8). For both cases, a traceless energy-momentum tensor is derived.

The question whether what one calls a "Carrollian theory" should be or not a theory invariant under local Carroll boosts (LCB) has been recently debated in the community. While authors in Refs. [22,28] argue that invariance under LCB naturally follows from a zero speed of light limit (namely a Carrollian or ultra-local limit) of Lorentzian theories, others (see e.g. in Ref. [24,26,46]) have been advocating for a breaking of LCB invariance. This discussion is nicely summarized in the manuscript in section 2.6 and presents some arguments in favor of keeping LCB.

The topic (construction of Carrollian action) and questions addressed are very timely and of interest for the community since, while they are expected to play an important role in flat space holography, very little is known so far about Carrollian field theories. Moreover, the paper is clear and well written; I am therefore happy to recommend it for publication in SciPost.

Requested changes

These are minor changes: 1 - It would be good to explain the notation for future null infinity in the second paragraph of the introduction. 2- In the fifth paragraph of the introduction, the sentence "Without these local boost symmetries, the notion of energy-momentum tensor appears to be problematic [26]" might sound misleading as it is my understanding that Ref. [26] argues in favor of not requiring local boost symmetry invariance. 3- It the last paragraph of the introduction, the "modified Mellin transform" is argued for only in Ref. [18], it is not involved in the proposal of Ref. [17]. 4- It the last paragraph of the introduction, one could consider adding Ref. arXiv:1707.09900 [hep-th] alongside with [35-39].

  • validity: -
  • significance: -
  • originality: -
  • clarity: -
  • formatting: -
  • grammar: -

Author:  Benjamin Søgaard  on 2022-12-16  [id 3140]

(in reply to Report 1 on 2022-10-28)

We thank the referee for their useful comments.

We have made the following changes in our resubmission to address the questions and requests of the referee: 1. Since we do not use it in the rest of the paper, we have just written out the notation $\mathcal{I}^\pm$ as "null infinity"in the second paragraph of the Introduction. 2. We now refer to reference [26] in a new sentence after the one mentioned in the fifth paragraph of the Introduction. This new sentence should better reflect the results obtained in [26], rather than our interpretation of these results. 3. We have separated the references to [17] and [18] in the last non-organizational paragraph of the Introduction and reworded it to properly reflect the difference between the transformations in both prescriptions. 4. The suggested reference has been added to the last paragraph of the Introduction.

---

## Round 2 · Referee Report · Anonymous (Referee 2) · 2022-11-28

Strengths

  1. This is a paper on Carrollian field theories that have gained a lot of recent attention particularly in the context of holography in asymptotically flat spacetimes. So the paper is on a subject of considerable current interest.

  2. It is a well written paper and is mathematically sound.

  3. The final part of the paper, where connections are made between Carrollian theories and Euclidean CFTs is particularly intriguing.

Weaknesses

Nothing in particular

Report

The paper under review revisits the formulation of Carrollian scalar field theories. Carrollian (quantum) field theories, especially Carrollian CFTs are of substantial recent interest due to potential connections to holography in asymptotically flat spacetimes, where they have been proposed as potential holographic duals.

The authors construct covariant Carrollian scalar field theories which have local boost as well as Weyl invariance in arbitrary dimensions. They find they are able to construct two distinct scalar theories, one which they call time-like and the other space-like. This in itself is not new and has been previously constructed in literature. However, I find the discussion in the present paper well organised and good to read. There is also a nice section in the geometric discussion after the introduction which compares different approaches in the literature. This should help people who are new to the field.

The section of the time-like action is useful background. But the meat of the paper is the section that follows on the space-like action. The author construct the action and check for its symmetries. The best part of the paper is Sec 4.3 where the authors explicitly show that by integrating out constants the space-like Carrollian theory can be reduced to a Euclidean relativistic CFT in one lower dimension. This was perhaps expected given the correlation functions of this theory look like CFT correlations in one lower dimension, but the explicit demonstration is impressive. I especially find the rewriting of Eq (4.37) to Eq (4.39) very nice.

I believe that the paper deserves publication in SciPost after a few things that need clarification first. I list my questions below.

  1. The authors say that (2.9) is not a boost invariant connection. Are quantities that arise out of this, e.g. the Ricci scalar, boost invariant?

  2. This is a general question which I am very confused with throughout the paper. Weyl invariance of the action automatically implies tracelessness of the EM tensor in a usual relativistic theory without imposing equations of motion. Why do the authors need to impose EOM to prove tracelessness in the Carrollian case? This does not seem correct.

  3. My main question is regarding Sec 4.3. The authors make the choice of gauge $b_i=0$. Although this is supposed to be a gauge choice, the various manipulations that follow depend crucially on this choice. I am worried that this may be a singular gauge choice and the rest of the construction would not work for arbitrary $b_i$ would not work. Can the authors work with an arbitrary $b_i$ and prove the claims later?

  4. Some minor typos throughout, including the rather interesting "witch" below Eq (4.32).

  • validity: -
  • significance: -
  • originality: -
  • clarity: -
  • formatting: -
  • grammar: -

Author:  Benjamin Søgaard  on 2022-12-16  [id 3141]

(in reply to Report 2 on 2022-11-28)

We are grateful to the referee for their constructive comments and remarks. One first point that we want to emphasize is that both our timelike and spacelike actions are new and have not previously appeared in the literature with conformal coupling to general Carroll backgrounds in general dimensions. Several references have considered the two-dimensional case, but there the conformal couplings vanish. The references [24,25] have constructed similar actions with conformal couplings in arbitrary dimensions, but in contrast to ours, these actions are not invariant under local Carroll boosts, as we emphasized in our Introduciton.

Next, we have modified our draft in response to the numbered questions as follows: 1. For general Carrollian backgrounds, most individual quantities derived from the connection (2.9) will not be invariant under Carroll boosts. In the way we perform the expansion, starting from actions that are invariant under Lorentz boosts, the results are guaranteed to be boost-invariant if everything is done correctly. For the spatial action, this boost-invariance is not manifest, and we explicitly check it in Section 4.1. We have added a remark below Equation (2.9) further stressing this point. 2. On this point we do not agree with the referee. Even if the action in question is Weyl-invariant, the energy-momentum tensor defined using the metric variation of a matter action will only be traceless off-shell if the matter fields have zero Weyl weight. This holds both in the conformal Carroll actions we construct and in the standard Lorentzian conformal scalar action (3.1). In this sense the fact that our Carroll energy-momentum tensors are only traceless on-shell in general dimension is consistent with our expectations. Additionally, already in the case of the Lorentzian conformal scalar action, we are not aware of improvement terms that can render the energy-momentum tensor traceless off-shell. 3. We thank the referee for pointing out this shortcoming in our previous presentation. Based on this query, we have significantly expanded Section 4.3. At the start of this section, specifically on page 23 and 24, we now give more explicit steps in the reduction of the spacelike action in the case where $b_i=0$. Additionally, an appropriate coordinate transformation can turn off $b_i$ on a given spatial slice even if it is initially nonzero. We discuss this on page 26 and show that the result of this reduction of the action is invariant under such a coordinate transformation, which corresponds to a spatial Weyl transformation of the fields in the reduced theory. 4. We have fixed this typo and carefully went over the draft another time.

---

## Round 3 · Referee Report · Anonymous (Referee 3) · 2022-12-20

Report

The authors have addressed my questions and concerns adequately. I recommend that the paper be published.

---

## Round 3 · Author Response

We have addressed the referee reports as individual replies.

---

## Round 3 · List of Changes

In addition to the changes mentioned in our replies to the reports, we have fixed a typo in Equation (2.8) and we slightly improved our wording in a few places.

---

## Editorial Decision

published